# FMRP has a cell-type-specific role in CA1 pyramidal neurons to regulate autism-related transcripts and circadian memory

Kirsty Sawicka[1‡]*, Caryn R Hale[1], Christopher Y Park[1§], John J Fak[1], Jodi E Gresack[2], Sarah J Van Driesche[1#], Jin Joo Kang[1], Jennifer C Darnell[1†¶]*, Robert B Darnell[1,3†]*

[1]Laboratory of Molecular Neuro-Oncology, The Rockefeller University, New York, United States; [2]Laboratory of Molecular and Cellular Neuroscience, The Rockefeller University, New York, United States; [3]Howard Hughes Medical Institute, Chevy Chase, United States

*For correspondence:
ksawicka@rockefeller.edu (KS);
darneje@rockefeller.edu (JCD);
darnelr@rockefeller.edu (RBD)

[†]These authors contributed equally to this work

Present address: [‡]Cancer Research UK Cambridge Institute, Li Ka Shing Centre, University of Cambridge, Cambridge, United Kingdom; [§]Flatiron Institute, New York, United States; [#]Amgen, Thousand Oaks, United States; [¶]Simons Initiative for the Developing Brain, University of Edinburgh, Edinburgh, United Kingdom

Competing interests: The authors declare that no competing interests exist.

**Abstract** Loss of the RNA binding protein FMRP causes Fragile X Syndrome (FXS), the most common cause of inherited intellectual disability, yet it is unknown how FMRP function varies across brain regions and cell types and how this contributes to disease pathophysiology. Here we use conditional tagging of FMRP and CLIP (FMRP cTag CLIP) to examine FMRP mRNA targets in hippocampal CA1 pyramidal neurons, a critical cell type for learning and memory relevant to FXS phenotypes. Integrating these data with analysis of ribosome-bound transcripts in these neurons revealed CA1-enriched binding of autism-relevant mRNAs, and CA1-specific regulation of transcripts encoding circadian proteins. This contrasted with different targets in cerebellar granule neurons, and was consistent with circadian defects in hippocampus-dependent memory in *Fmr1* knockout mice. These findings demonstrate differential FMRP-dependent regulation of mRNAs across neuronal cell types that may contribute to phenotypes such as memory defects and sleep disturbance associated with FXS.

## Introduction

Fragile X Syndrome (FXS), the most common inherited cause of intellectual disability and leading monogenic cause of autism, results from loss of function of the RNA binding protein FMRP (*Hagerman et al., 2017*). This loss of function is typically due to transcriptional silencing of the *Fmr1* gene (*Pieretti et al., 1991*), although missense mutations in either of the KH-type RNA binding domains, frame shift mutations and deletions can also cause the phenotype in human (*Coffee et al., 2008*; *De Boulle et al., 1993*; *Deciphering Developmental Disorders Study, 2015*; *Hammond et al., 1997*; *Lugenbeel et al., 1995*; *Myrick et al., 2014*; *Quartier et al., 2017*; *Sitzmann et al., 2018*; *Suhl and Warren, 2015*) and rodent models (*Till et al., 2015*; *Zang et al., 2009*). Individuals with FXS suffer from a range of cognitive and behavioral deficits that can include social deficits, anxiety, stereotypic movements, hyperactivity, seizures, memory deficits, and sleep dysfunction (*Kronk et al., 2010*; *Munir et al., 2000*; *Ornstein et al., 2008*; *Richdale, 2003*; *Wadell et al., 2013*) and approximately 50% of males with FXS meet the diagnostic criteria for autism (*Clifford et al., 2007*; *Hall et al., 2008*; *Harris et al., 2008*). The *Fmr1* knockout (KO) mouse, or *Fmr1* knock-in mouse harboring a I304N missense mutation (*Zang et al., 2009*) seen in a severely affected FXS individual (*De Boulle et al., 1993*), exhibit a range of neurologic, behavioral and cognitive deficits, including defects in neuronal circuits involved in hippocampal memory (*Arbab et al., 2018*; *Boone et al., 2018*; *Huber et al., 2002*; *Talbot et al., 2018*) and in circadian rhythm

(*Zhang et al., 2008*), as well as altered dendritic spine morphology and dynamics, synaptic plasticity and neuronal circuits that model findings made in humans (*Kazdoba et al., 2014*).

Given the critical role of FMRP in brain function, the transcripts it binds and regulates has been an area of great interest and is a cornerstone to understanding the pathophysiology of FXS. Early in vitro experiments led to identification of high affinity targets (*Darnell et al., 2005*; *Darnell et al., 2001*) and subsequent in vivo studies from whole mouse brain revealed robust FMRP binding to a discrete subset of mRNAs (*Brown et al., 2001*; *Darnell et al., 2011*). The identified FMRP targets were enriched in mRNAs encoding synaptic and chromatin regulatory proteins, and significantly overlapped with autism susceptibility genes identified through genetic studies (*Darnell et al., 2011*; *Iossifov et al., 2012*; *Purcell et al., 2014*; *Zhou et al., 2019*).

There is increasing evidence that there are brain region and cell-type specific deficits in the absence of FMRP. For example, *Fmr1* KO mice exhibit synapse- and region-specific defects in synaptic plasticity, inhibitory and excitatory circuits (*Dahlhaus, 2018*), dendritic spine morphology and dynamics (*He and Portera-Cailliau, 2013*) and mTOR and MAPK signaling pathways (*Sawicka et al., 2016*). Genetic studies using the *Fmr1* cKO mouse have also demonstrated that specific cell populations contribute to specific deficits in cerebellar function (*Guo et al., 2011*; *Koekkoek et al., 2005*; *Mientjes et al., 2006*). However, it is as yet unknown if FMRP binding or function varies across cell types.

We recently developed a means of assessing regulatory protein-RNA interactions in specific cell types in vivo. This method, termed cTag CLIP (*Hwang et al., 2017*; *Saito et al., 2018*; *Ule et al., 2018*), combines CLIP (UV cross-linking immunoprecipitation) technology with the specificity that can be conferred by Cre-lox technology. RNA binding proteins are modified by homologous recombination to allow Cre-dependent knock-ins of epitope tags that are then used for CLIP purification. This cell-type specific Cre-lox mediated switching of epitope tags allows for minimal perturbation of the stoichiometry and regulation of the RNA binding protein of interest, a key factor in ascertaining biologically relevant RNA-protein interactions in vivo. Using antibodies to epitope tags confers uniformity on CLIP protein-RNA purification, while allowing cell-type specific RNA-protein complexes to be extracted from complex brain tissue.

For example, cTag CLIP applied to the ubiquitous poly(A) binding protein PABPC1 (*Hwang et al., 2017*; *Hwang and Darnell, 2017*; *Jereb et al., 2018*) allowed dissection of different PABPC1-bound poly(A) mRNAs from cortical excitatory neurons, inhibitory neurons, microglia and astrocytes, and in the cerebellum, from excitatory granule neurons and inhibitory Purkinje neurons (*Jereb et al., 2018*). More recently, we have also applied cTag CLIP to discern differential regulation of the same transcripts by the splicing factor, NOVA2, in excitatory versus inhibitory cortical and cerebellar neurons (*Saito et al., 2018*).

Given the interest in FMRP cell-specific function in specific brain regions and cell types, we developed cTag FMRP CLIP in CA1 pyramidal neurons of the mouse hippocampus, a cell type which plays a critically important role in long-term memory and spatial related tasks. The *Fmr1* KO mouse exhibits abnormal synaptic plasticity and spine morphology in this cell type which has been linked to altered hippocampal function and defects in learning and memory (*Arbab et al., 2018*; *Ceolin et al., 2017*; *Grossman et al., 2006*; *Huber et al., 2002*; *Lauterborn et al., 2007*; *Nosyreva and Huber, 2006*; *Talbot et al., 2018*; *Thomson et al., 2017*). Importantly, we have combined these studies with cell-type specific assessment of ribosome bound transcripts in this same cell type, to allow a comprehensive, unbiased and normalized approach to assessing FMRP-bound transcripts. Furthermore, comparative analysis of FMRP binding and regulation in hippocampal CA1 versus cerebellar excitatory neurons revealed CA1-specific enrichment of binding to transcripts related to Autism Spectrum Disorders (ASD). Unexpectedly, these analyses also revealed FMRP-dependent regulation of mRNAs encoding circadian proteins in hippocampal CA1, but not cerebellar granule cells, and that loss of FMRP results in circadian-dependent defects in learning and memory.

## Results

### FMRP cTag CLIP in CA1 neurons

To determine FMRP-binding maps in specific cell types, we generated *Fmr1*-cTag mice, which enable conditional expression of AcGFP (Aequorea coerulescens Green Fluorescent Protein) tagged FMRP using the Cre-lox system (*Van Driesche et al., 2019*). A knock-in strategy was employed, targeting the endogenous *Fmr1* locus, in order to maintain wild-type gene expression and regulation (*Figure 1A*). The addition of the AcGFP-tag enables capture of all major isoforms of FMRP with the exception of a small minority of nuclear isoforms with an alternative c-terminus resulting from exclusion of exon 14. Nearly identical CLIP results have been observed with AcGFP-tagged FMRP and endogenous FMRP in forebrain neurons validating the use of the *Fmr1*-cTag mouse for CLIP studies (see Figure 2 in *Van Driesche et al., 2019*) and a similar strategy has been successfully implemented for other RNA-binding proteins including NOVA2 (*Saito et al., 2018*) and PABPC1 (*Hwang et al., 2017*; *Hwang and Darnell, 2017*; *Jereb et al., 2018*).

To achieve CA1-restricted expression of GFP-tagged FMRP, we crossed *Fmr1*-cTag mice with the *Camk2a-Cre* mouse line in which the mouse *Camk2a* promoter drives Cre recombinase expression in a subset of forebrain excitatory neurons and, within the hippocampus, specifically in CA1 pyramidal neurons (*Tsien et al., 1996*). Immunofluorescence of hippocampal tissue from *Camk2a-Cre*[+/-]; *Fmr1*-cTag[+/Y] (*Fmr1*-cTag[Camk2a-Cre]) confirmed clear expression of GFP in CA1 pyramidal cells with no detectable expression in the absence of Cre recombinase (*Figure 1B*). Within the CA1 pyramidal cell expression of AcGFP-tagged FMRP was highest in the cell soma with detectable expression in the proximal dendrites (*Figure 1B*). This expression pattern is consistent with that observed for endogenous FMRP albeit only in a subset of pyramidal neurons (*Figure 1—figure supplement 1*), consistent with the level of activity of the Camk2a Cre driver at this developmental age.

To precisely identify FMRP-RNA interactions within hippocampal CA1 pyramidal neurons, we performed CLIP from 3 cohorts of *Fmr1*-cTag[Camk2a-Cre] male mice aged postnatal day 28–32 and Cre-negative controls. Using anti-GFP antibodies, directly bound RNA fragments UV cross-linked to FMRP-GFP specifically in CA1 neurons were isolated, cloned and sequenced. These sequenced RNA 'tags' were mapped to the transcriptome to identify sites of FMRP binding (*Supplementary file 1*). Results from three biological replicates show good correlation for the numbers of mapped tags per transcript (*Figure 1C*) and tag distribution (*Figure 1D*). Overall, 86% of reads mapped to annotated genes and of these 62% mapped within the coding region. Binding was also seen within 3'UTR regions, but decreased across the proximal 500nt of the 3'UTR to approximately 50% of the coverage seen within the coding region (*Figure 1E*). Each FMRP target has its own unique CLIP read distribution and the majority show continuous coverage across the coding region (*Figure 1F*).

### Identification of FMRP targets

Aside from the ability of single cell type cTag CLIP to identify binding events from a specific cell population, it has the added advantage that the CLIP reads can be normalized for transcript abundance in that cell type and FMRP targets thereby ranked by their relative binding. We used TRAP (Translating Ribosome Affinity Purification) to isolate cell-type-specific ribosome-associated mRNAs (*Doyle et al., 2008*; *Heiman et al., 2008*). This has the advantage that it does not require physical isolation of the cell type of interest which can alter the expression profile of the cell and results in loss of dendritic processes. Furthermore, although we recognize that it limits quantitation to ribosome-bound mRNAs, since FMRP is known to be predominantly associated with polyribosomes in the brain (*Darnell et al., 2011*) TRAP seems suitable for normalizing CA1 FMRP CLIP tags to CA1 cellular transcript abundance.

We used the RiboTag mouse (*Sanz et al., 2009*), which enables conditional HA tagging of the endogenous RPL22 ribosomal protein, and crossed these mice with the same *Camk2a-Cre* line to specifically label ribosomes with a triple HA tag in CA1 pyramidal cells. Immunoprecipitation with an HA antibody, in the presence of cycloheximide to maintain ribosome-mRNA association, enables specific isolation of ribosome-bound RNA from the cell type of interest. We optimized the concentration of HA antibody for TRAP (*Figure 2—figure supplement 1*), and validated the method by examining the enrichment of mRNAs encoding CA1 specific markers and the depletion of markers

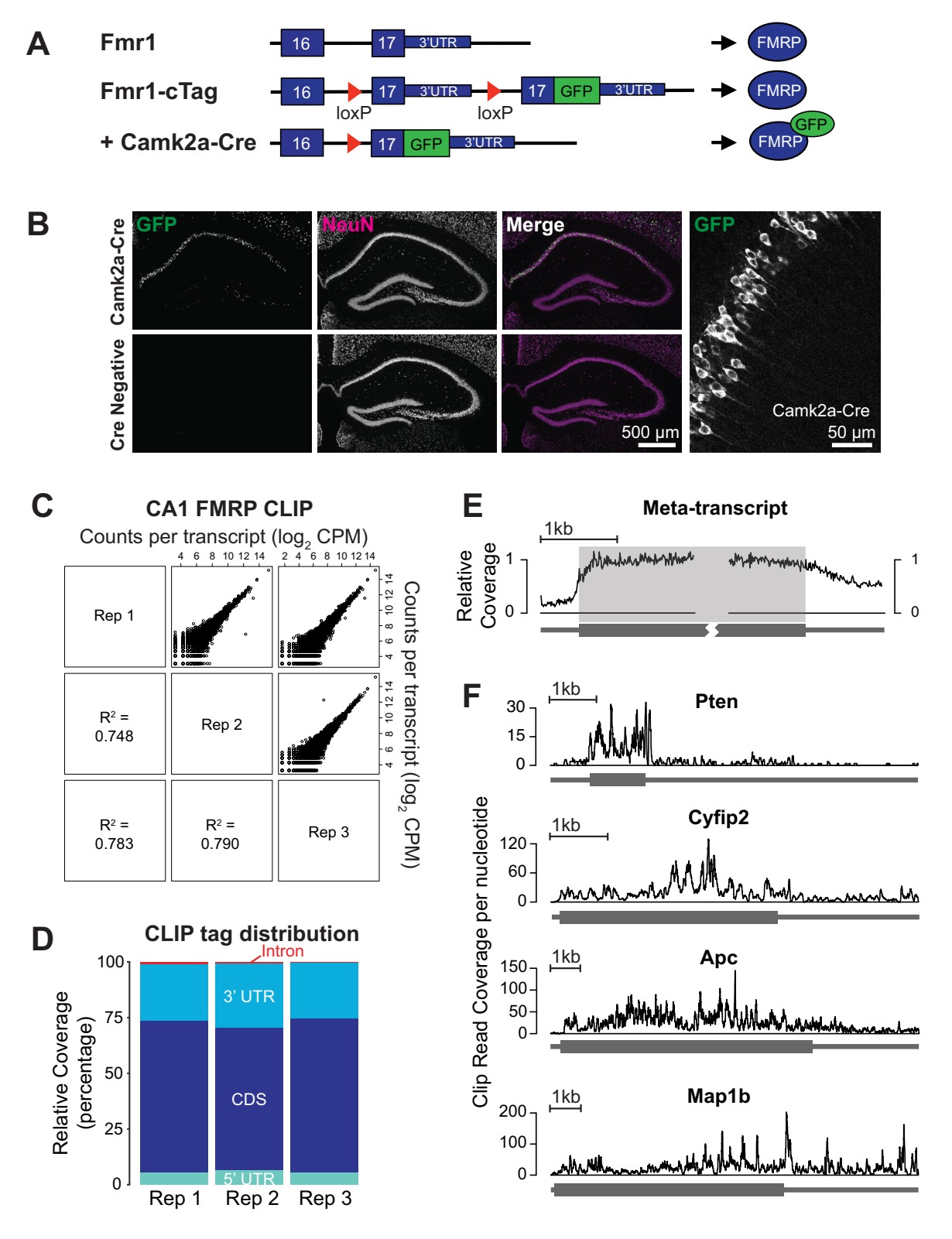

**Figure 1.** CA1 pyramidal neuron specific FMRP CLIP. (**A**) Schematic of the *Fmr1*-cTag conditional knock-in mouse. The final exon of the *Fmr1* gene is flanked by loxP sites and a second copy of this exon with an additional AcGFP sequence at the end of the coding region is cloned downstream. In the absence of Cre-dependent recombination the endogenous untagged protein is expressed. Cre-dependent recombination causes usage of the alternative final exon and expression of the GFP-tagged FMRP. (**B**) The *Fmr1*-cTag mouse was crossed with a *Camk2a-Cre* mouse line. Cre-dependent

*Figure 1 continued*

recombination and expression of the GFP-tagged FMRP occurs predominantly in the CA1 region. Brain sections were prepared from P29 *Fmr1-*cTag[Camk2a-Cre] mice (Camk2a-Cre) or *Fmr1-*cTag mice (Cre Negative). Immunofluorescence was performed using a GFP antibody and an antibody to the neuronal marker NeuN, shown in green and magenta respectively in merged image. Higher magnification image of the *Fmr1-*cTag[Camk2a-Cre] mice GFP staining shows presence of the GFP-tagged FMRP in the cytoplasm and proximal dendrites of the CA1 neurons. (**C**) Three biological replicates of CA1 FMRP cTag CLIP from P28-32 male *Fmr1-*cTag[Camk2a-Cre] mice are well correlated. The scatter plots of read counts per million per transcript (CPM) for each pairwise comparison are shown. Numbers represent Pearson $R^2$ correlation between the replicates. (**D**) CLIP read distribution across genic regions from three biological replicates shows that FMRP predominantly binds the coding sequence with some binding within 3'UTRs. Counts of reads mapping to each region were normalized for the overall sequence length of each feature. (**E**) Meta-transcript coverage profile showing the relative CLIP read coverage upstream and downstream of the start and stop codons. Coverage was calculated from the 1000 transcripts with highest CLIP tag density and plotted relative to the coding region. The coding region is highlighted in gray. (**F**) Coverage plots for four well-established FMRP targets showing the total number of CLIP reads from all three replicates that overlap each nucleotide in the transcript.

The online version of this article includes the following figure supplement(s) for figure 1:

**Figure supplement 1.** Expression of AcGFP-tagged FMRP protein.

for other cell types in our immunoprecipitation relative to the input lysate (*Figure 2A*, *Figure 2—figure supplement 2A*).

We further validated our TRAP data as a suitable measure of transcript abundance by comparing it to RNA-Seq from isolated CA1 pyramidal cells. A conditional tdTomato reporter mouse line was crossed with the same *Camk2a-Cre* line used for TRAP and CLIP experiments to specifically label the cell population of interest with a fluorescent protein. Hippocampal cells from these animals were dissociated and the tdTomato-positive CA1 neurons were identified and isolated by fluorescence-activated cell sorting (FACS). After sequencing, the RNA obtained by either TRAP or FACS showed good correlation (*Figure 2—figure supplement 2B*; Pearson correlation, $R^2 = 0.87$) confirming that TRAP efficiently isolates the majority of the transcriptome.

We sought to determine an appropriate metric to define the propensity for FMRP to bind to each individual mRNA based on crosslinking events identified by CLIP and its abundance as defined by TRAP. We focused on interactions within the coding region since this was the most consistent region of binding across different transcripts and also the region of binding most clearly linked to FMRP function (*Darnell et al., 2011*). CLIP tag density within the coding region was significantly correlated with the abundance of the transcript as measured by TRAP (*Figure 2B*, Spearman correlation test $p < 2.2 \times 10^{-16}$, $R^2 = 0.31$). We made use of this observation to calculate a 'CLIP score' for each transcript that defines the amount of FMRP binding relative to other transcripts of a similar abundance. For each CLIP biological replicate a linear regression line was fitted through a plot of $\log_2$ CLIP RPKM *vs* $\log_2$ TRAP RPKM and the position of each transcript was calculated relative to this fitted line (*Figure 2—figure supplement 3A*). The further above the regression line a transcript fell the higher the CLIP score and the greater the amount of FMRP binding was inferred.

Using this calculated CLIP score from all three replicates, we then classified each transcript by the extent of FMRP binding. 327 stringent FMRP targets were empirically defined as transcripts with a CLIP score >2 in all three replicates; 938 high binding targets had a mean CLIP score >1; 1330 low binding targets had a mean CLIP score between 0 and 1; all other transcripts were classed as non-targets (8288 transcripts) with either lower CLIP scores (6565 transcripts) or no CLIP tags (1723 transcripts; *Figure 2B*, *Supplementary file 2*).

We additionally compared the CLIP Score metric described above to that calculated by two other variants of this method. Firstly, we substituted TRAP RPKM as a measure of CA1 neuron transcript abundance for RPKM values calculated from FACS RNA-Seq and used the same linear regression method to calculate a CLIP Score for each transcript. Secondly, we used raw counts rather than RPKM values to compare CLIP and TRAP data. This alternative count-based approach incorporated a dispersion estimate and used a negative binomial distribution to determine significance. CLIP scores were determined as before based on residuals from a linear regression model. In addition, transcripts were assigned a p-value based on the significance of deviation of the observed CLIP count compared to the expected count. The calculated CLIP scores were very similar across all three methods (*Figure 2—figure supplement 4*, *Supplementary file 5*). We chose to use the linear regression model with TRAP RPKM as TRAP can better capture the in vivo transcriptome including

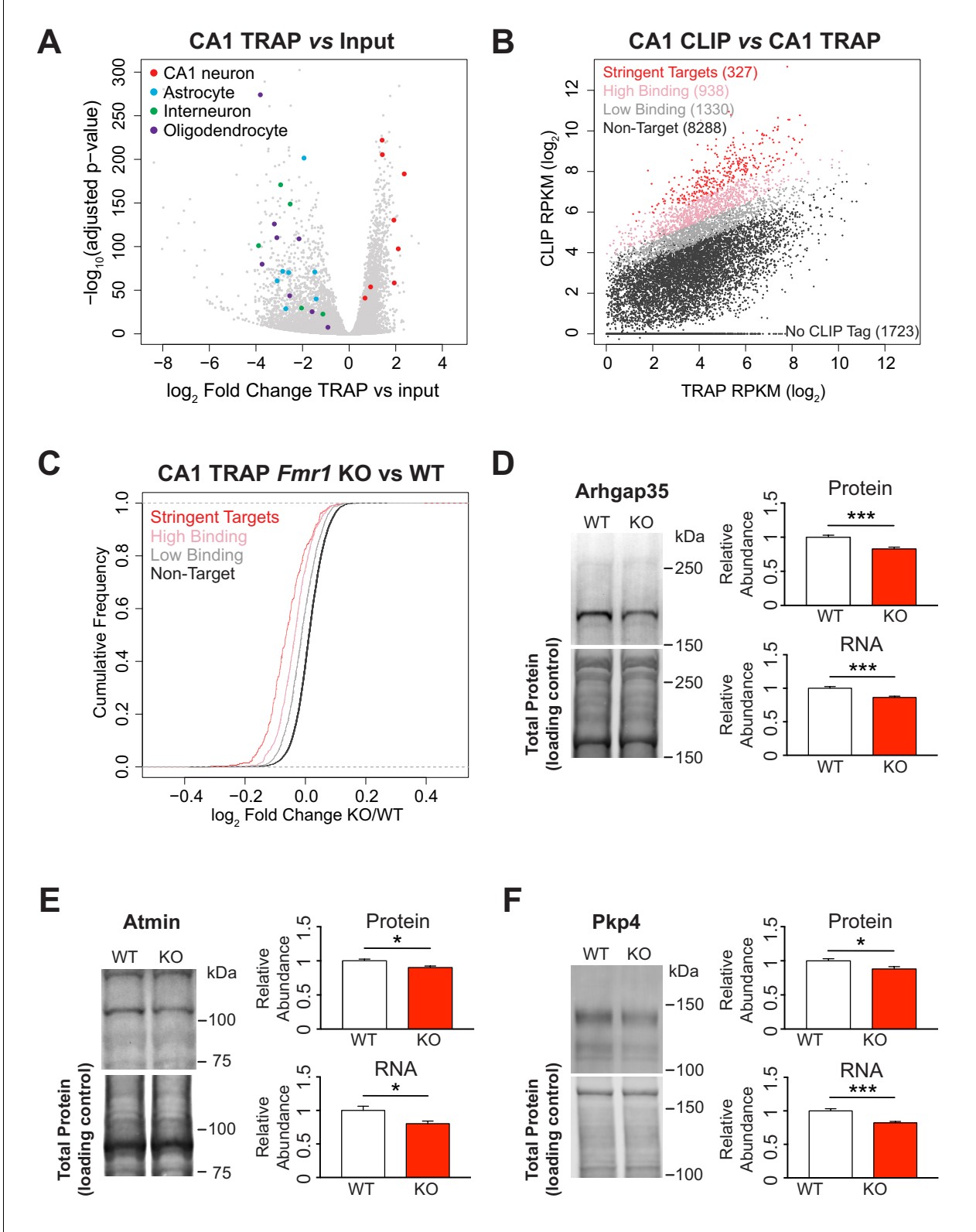

**Figure 2.** Identification of FMRP targets and CA1 neuron specific TRAP in the *Fmr1* KO. (**A**) Immunoprecipitation of ribosome associated RNA from CA1 neurons from WT and *Fmr1* KO animals. Differential analysis of counts per gene from TRAP RNA compared to input RNA showing fold change and statistical significance as determined by DESeq2. CA1 markers (Wfs1, Pou3f1, Nov, Man1a, Mpped1, Cds1, Fibcd1, Satb2) are enriched in the IP whereas markers for other cell types are depleted (Astrocytes: Slc1a2, Gjb6, Gfap, Slc1a3, Aqp4, Aldoc, Aldh1l1; Interneurons: Slc6a1, Slc32a1, Gad1,

*Figure 2 continued on next page*

*Figure 2 continued*

Gad2, Pvalb; Oligodendrocytes: Cnp, Mog, Mag, Olig2, Olig1, Mobp, Mbp, Mal). (**B**) FMRP targets were defined by normalizing CLIP tag density across the coding region to the relative abundance of the transcript as measured by TRAP. A CLIP score per transcript was calculated independently for each replicate of the CLIP experiment. Stringent targets were defined as those with a CLIP score > 2 in every replicate, targets with high binding were defined as those with an average CLIP score > 1 and targets with low binding were defined as those with an average CLIP score between 0 and 1. Scatter plot of average density of CLIP tags across the coding region (CLIP RPKM) *vs* transcript abundance calculated as TRAP read density across the full transcript (TRAP RPKM). Targets of each subclass are highlighted in the plot and the number of genes within each subclass is indicated. (**C**) FMRP targets are down-regulated in the *Fmr1* KO, with the magnitude of the effect being proportional to the amount of FMRP binding. Cumulative density function plots of the $\log_2$ fold change between *Fmr1* KO TRAP and WT TRAP for each FMRP target subclass. All subclasses have a significant shift compared to the unbound group (Kolmogorov-Smirnov test, p-value<$2.2 \times 10^{-16}$ for all pairwise comparisons) and there are also significant differences between each subclass (stringent binding vs high binding: p-value=$5.78 \times 10^{-9}$; high binding vs low binding: p-value<$2.2 \times 10^{-16}$). (**D–F**) Quantitative western blot and PCR validation of TRAP results. Protein and RNA levels of the FMRP targets Arhgap35 (**D**), Atmin (**E**) and Pkp4 (**F**) are decreased in hippocampal lysates from *Fmr1* KO mice relative to WT. A representative western blot is shown for one pair of WT and KO littermates. Western blots were normalized to total protein per lane visualized with REVERT total protein stain. qPCR data was normalized to Gapdh, Actin and Hprt housekeeping genes. Data are from 6 to 9 animals per genotype and mean ± SEM is shown. p-values were calculated using Student's t-test (*p<0.05, ***p<0.001).

The online version of this article includes the following figure supplement(s) for figure 2:

**Figure supplement 1.** Optimization of TRAP immunoprecipitation.
**Figure supplement 2.** Validation of TRAP method.
**Figure supplement 3.** Identification of FMRP targets and comparison with other published data and cell types.
**Figure supplement 4.** Comparison of different CLIP score methods.

dendritic mRNAs with minimum perturbation of the cells and use of RPKM should help minimize any potential length bias in our data.

## Most FMRP targets are down-regulated in the *Fmr1* KO

To examine whether the FMRP CLIP score metric we defined correlates with the functional consequence of loss of FMRP, we performed TRAP from the same CA1 neurons in the *Fmr1* KO (*Supplementary file 3*). We observed a significant decrease in FMRP target transcripts associated with ribosomes in *Fmr1* KO CA1 neurons (*Figure 2C*; Kolmogorov-Smirnov test, p-value<$2.2 \times 10{-6}$). Transcripts that showed the most binding in WT mice (stringent FMRP targets) showed the greatest decrease in the absence of FMRP. Similarly, high binding and low binding targets were also significantly decreased, with the effect proportional to the amount of FMRP normally bound. The same decrease in FMRP targets was also observed among targets defined by our two alternative normalization methods further confirming that all three methods successfully identify a population of functionally-relevant FMRP-bound transcripts (*Figure 2—figure supplement 4C*).

Our results are consistent with those of others who have reported a decrease of FMRP targets in CA1 TRAP in the Fragile X mouse (*Ceolin et al., 2017*; *Thomson et al., 2017*). For example, Ceolin et al. found 15% of differentially regulated genes in the *Fmr1* KO were FMRP target mRNAs and of these 11/12 FMRP target mRNAs were downregulated in hippocampal neurons. Reanalysis of their data in relation to our CA1 cTag CLIP-defined FMRP targets showed remarkably similar results to our own despite use of a different CA1-specific Cre driver and an older cohort of animals (*Figure 2— figure supplement 3B*, *Supplementary file 3*), independently supporting our data and indicating that down-regulation of FMRP target mRNAs persists into adulthood.

To determine whether down-regulation of FMRP targets occurs in another cell type, we considered cerebellar granule cells, another excitatory neuronal cell type for which FMRP cTag CLIP was available (*Van Driesche et al., 2019*). As for the CA1 CLIP data, we defined 3 classes of FMRP targets based on their CLIP scores (*Figure 2—figure supplement 3C*). We additionally performed TRAP from WT and *Fmr1* KO for granule neurons using the same *NeuroD1-Cre* mouse line with granule cell specific expression within the cerebellum as for the original CLIP studies. As in CA1 neurons, FMRP targets were down-regulated in granule neurons in absence of FMRP with the decrease being proportional to the amount of binding in WT animals (*Figure 2—figure supplement 3D*, *Supplementary file 3*).

A change in the levels of an mRNA in the TRAP-captured pool in the *Fmr1* KO may indicate a decrease in transcript steady-state levels and/or a difference in the proportion of the transcript that

is ribosome associated. Either of these possibilities would most likely lead to a decrease in the synthesis of the encoded protein. To address this, we chose three FMRP targets that were among the most down-regulated transcripts by TRAP and for which antibodies were available, Arhgap35, Atmin and Pkp4. Western blots from whole hippocampal lysates for all three proteins showed a significant decrease in *Fmr1* KO mice relative to their WT littermates, in agreement with the TRAP results (*Figure 2D–F*; Student's t-test: Arhgap35 p=0.008, Atmin p=0.016, Pkp4 p=0.022, n = 6–8 mice per genotype). qPCR from parallel tissue samples also showed a similar decrease in RNA levels for these transcripts (*Figure 2D–F*; Student's t-test: Arhgap35 p=0.0005, Atmin p=0.016, Pkp4 p=0.0003, n = 8–9 mice per genotype) indicating that, at least for these targets, the decrease in the TRAP RNA was largely the result of a decrease in transcript abundance. This is consistent with recent findings in adult neural stem cells which showed a decrease in FMRP targets at the mRNA level by RNA-seq but little change in the overall number of associated ribosomes by ribosome profiling (*Liu et al., 2018*).

## FMRP actions in hippocampal versus cerebellar excitatory neurons

Comparison of our new CA1 target list (encompassing targets in the stringent and high binding target groups) with our latest list of whole brain FMRP targets, and targets determined by FMRP cTag CLIP and defined by the same parameters in cerebellar granule cells, revealed 246 transcripts that are uniquely identified in CA1 neurons (*Figure 3A*). Further analysis of FMRP cTag CLIP data from hippocampal CA1 and cerebellar excitatory neurons after normalization for transcript abundance, showed that CLIP scores defined in these two cell types were significantly correlated, indicating that binding affinity is broadly conserved between these cell types (Pearson correlation $R^2$ = 0.48, *Figure 3B*). However, statistical analysis of CLIP scores from three biological replicates for each cell type revealed 248 transcripts with differential binding, 148 preferentially bound in CA1 and 100 in granule cells (*Figure 3B*, *Supplementary file 3*). Many of the transcripts that are more highly bound in CA1 neurons are related to known Fragile X or autism phenotypes, including mRNAs encoding postsynaptic proteins such as Shank1, Shank3, Syngap1 and Psd95 (Dlg4). SHANK3 is deleted in Phelan-McDermid syndrome, a disorder that is frequently comorbid with ASD and recurrent mutations in the SHANK3 gene have been identified in individuals with ASD. Multiple loss of function variants in SYNGAP1 have also been identified in patients with ASD and intellectual disability, making these two of the most well characterized autism-related genes. Elevated PI3K-mTOR signaling has been observed in the hippocampus of Fragile X mice (*Sharma et al., 2010*) and has been linked to elevated levels of PI3K Enhancer (PIKE) encoded by AGAP2, another autism candidate gene with enriched FMRP binding in CA1. PIKE activates PI3K downstream of group one metabotropic glutamate receptors and genetic reduction of this protein in *Fmr1* KO mice has been shown to rescue dendritic spine morphology in CA1 neurons and elevated mGluR-LTD at Schaeffer collateral-CA1 synapses (*Gross et al., 2015a*).

## FMRP binds long, autism-related mRNAs

To further explore the association of FMRP with autism-related mRNAs, we examined binding to the mRNAs encoded by SFARI autism candidate genes in CA1 neurons (*Figure 3C*). Strikingly, FMRP binding was highly enriched on autism candidate mRNAs and the amount of binding correlated with the strength of evidence supporting the classification. That is to say that the higher the probability of a gene being causative to ASD phenotypes, the more highly bound it is by FMRP in CA1 neurons (*Figure 3C*; Wilcoxon rank sum test for SFARI gene clip scores compared to clip scores for all CA1 genes: SFARI category 1, p=$8.1\times10^{-11}$, n = 25; SFARI category 2, p=$5.2\times10^{-13}$, n = 61, SFARI category 3, p=$2.3\times10^{-16}$, n = 184; SFARI category 4, p<$2.2\times10^{-16}$, n = 437; SFARI category 5, p=$3.5\times10^{-4}$, n = 170; SFARI category 6, p=0.71, n = 24; SFARI Syndromic, p<$2.2\times10^{-16}$, n = 166). Moreover, CA1 FMRP targets show significantly higher ASD gene scores than granule cell FMRP targets (*Figure 3D*; Wilcoxon rank sum test p=$1.2\times10^{-3}$) and transcripts with enriched binding in CA1 neurons have significantly higher ASD gene scores compared to those preferentially bound in granule cells (*Figure 3D*; Wilcoxon rank sum test p=$3.1\times10^{-8}$). A higher gene score in this analysis indicates a gene whose functional network neighborhood is significantly enriched for genes with stronger than predicted disease impact in ASD proband de novo mutations compared to unaffected siblings (*Zhou et al., 2019*). These observations strengthen and refine the relationship between

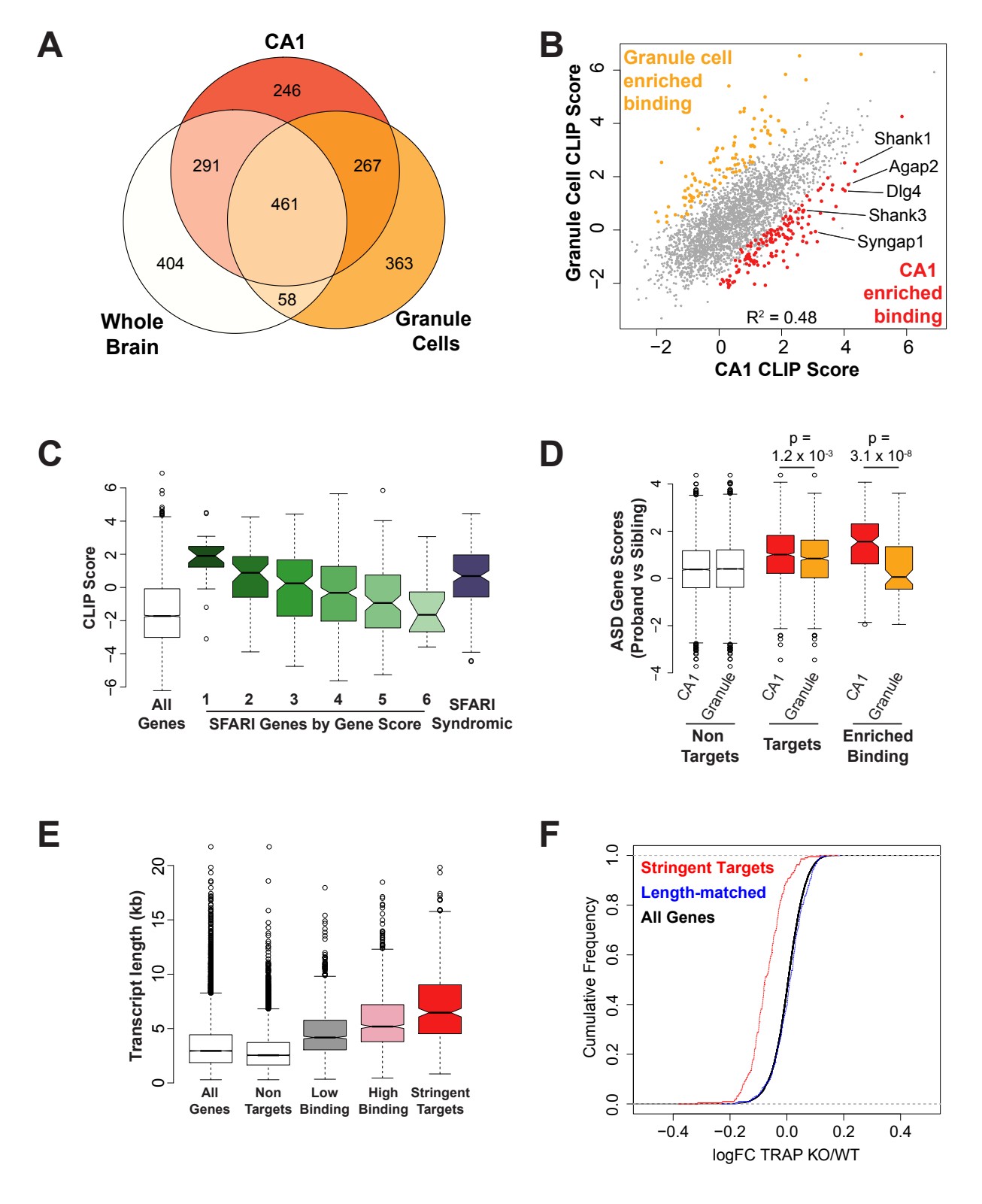

**Figure 3.** FMRP binds autism-related mRNAs and long transcripts. (**A**) Venn Diagram comparing FMRP targets in CA1 neurons to those determined from whole brain FMRP CLIP and cerebellar granule cell CLIP. Targets were defined as CLIP score >1 for CA1 and granule cells or as false discovery rate (FDR) < 0.05 and detection in at least 6 of 10 experiments for whole brain. (**B**) Scatter plot of FMRP CLIP score in CA1 neurons compared to FMRP CLIP score in cerebellar granule cells. Transcripts with significantly enriched binding in CA1 (p<0.05) are indicated in red. CA1-enriched targets relevant
*Figure 3 continued on next page*

*Figure 3 continued*

to Fragile X phenotypes are highlighted. (**C**) FMRP is highly associated with autism-related transcripts in CA1. SFARI genes associated with Autism Spectrum Disorders, were grouped according to their gene score where a lower gene score indicates a higher strength of the evidence linking candidate genes to ASD. Category 1: High confidence, Category 2: Strong candidate, Category 3: Suggestive evidence, Category 4: Minimal evidence, Category 5: Hypothesized but untested, Category 6: Evidence does not support a role, Syndromic: Genes predisposing to autism in the context of a syndromic disorder. Box and whisker plot of CLIP scores for genes in each class shows greatest association of FMRP with the highest confidence ASD gene candidates. Wilcoxon rank sum test for SFARI gene CLIP scores compared to CLIP scores for all CA1 expressed genes: SFARI category 1, p=$8.1\times10^{-11}$, n = 25; SFARI category 2, p=$5.2\times10^{-13}$, n = 61, SFARI category 3, p=$2.3\times10^{-16}$, n = 184; SFARI category 4, p<$2.2\times10^{-16}$, n = 437; SFARI category 5, p=$3.5\times10^{-4}$, n = 170; SFARI category 6, p=0.71, n = 24; SFARI Syndromic, p<$2.2\times10^{-16}$, n = 166. (**D**) FMRP binding to ASD-related transcripts is enriched in CA1 neurons relative to cerebellar granule cells. ASD gene scores were determined by Network-neighborhood Differential Enrichment Analysis of de novo mutations in ASD proband versus unaffected sibling (*Zhou et al., 2019*) such that genes with higher gene scores are part of a network or pathway predicted to have a significant disease impact. ASD gene scores were compared across FMRP non-targets and targets determined by CA1 and granule cell CLIP. Targets were defined as CLIP score >1. ASD gene scores were also compared between transcripts with enriched binding in either CA1 neurons or cerebellar granule cells. Only transcripts expressed in both cell types were included in the analysis. p-values were calculated by Wilcoxon rank sum test. (**E**) FMRP preferentially associates with long transcripts. FMRP targets of all classes consist of transcripts of significantly longer than average length with the effect being greatest for the most stringent targets. Stringent targets were defined as those with a CLIP score >2 in every replicate, targets with high binding were defined as those with an average CLIP score >1, targets with low binding were defined as those with an average CLIP score between 0 and 1 and non-targets were all transcripts with a CLIP score <0. All classes of FMRP targets were significantly longer than the full transcriptome (p-value<$2.2\times10^{-16}$ for all pairwise comparisons, Wilcoxon rank sum test). (**F**) FMRP regulates a specific subset of long transcripts. A set of FMRP stringent targets (CLIP score > 2 in every replicate) and non-targets (CLIP score < 1) were selected that had the same distribution of transcript lengths and TRAP RPKM in the range 10–200. Cumulative density function plots of the log2 fold change between *Fmr1* KO TRAP and WT TRAP for these subsets show specific regulation of the bound transcripts. Kolmogorov-Smirnov test: stringent targets vs length-matched non-targets, p-value<2.2×10–16.

The online version of this article includes the following figure supplement(s) for figure 3:

**Figure supplement 1.** Comparison of FMRP and MeCP2 targets and controls for length or expression bias.

FMRP-regulated transcripts and those implicated in ASD, demonstrating here a previously unrecognized cell-specific link between the two.

Long genes and transcripts are preferentially expressed in mouse and human brain relative to other tissues and such genes typically have brain specific function and expression. While many neuronal populations express longer transcripts relative to other cell types, CA1 pyramidal cells in particular express a greater number of long transcripts and have a transcriptome with significantly longer transcripts than that of somatosensory cortex pyramidal neurons (*Gabel et al., 2015*; *Zylka et al., 2015*). Consistent with the enrichment for neuronal function among long genes, dysregulated transcription of long genes has been implicated in autism, intellectual disability, Rett Syndrome and FXS (*Gabel et al., 2015*; *Greenblatt and Spradling, 2018*; *King et al., 2013*). To determine whether FMRP binding is enriched on long transcripts in CA1 neurons, we examined the length of transcripts in our subclasses of targets. FMRP binding is highly enriched on long transcripts (length of FMRP targets vs CA1 expressed transcripts, Wilcoxon rank sum test, p<2.2×10–6) and transcript length increases for more highly bound targets (*Figure 3E*).

Given the existing evidence that mutations in MeCP2, the underlying cause of Rett Syndrome, may cause neurological dysfunction by specifically disrupting long gene expression in the brain, and previously reported overlaps between FMRP and MeCP2 targets, we tested the overlap of our CA1 FMRP target list with a set of genes that are transcriptionally repressed by MeCP2 (*Gabel et al., 2015*). A significant overlap was observed between FMRP and MeCP2 targets (*Figure 3—figure supplement 1A*; Fishers exact test, p=$6.94\times10^{-6}$) and MeCP2 targets were found to have a small but significant decrease in TRAP from *Fmr1* KO mice (*Figure 3—figure supplement 1B*, Kolmogorov-Smirnov test, p<$2.2\times10^{-16}$). However, only a minority of MeCP2-repressed genes are FMRP targets, with 20.7% having a CLIP score >1 and 5.3% being classed as stringent FMRP targets.

To further explore the link between FMRP binding and transcript length, we compared FMRP targets to a random set of length-matched transcripts with either low or no FMRP binding. The length-matched set of transcripts showed no dysregulation in the *Fmr1* KO (*Figure 3F*), suggesting that length is not the sole determining factor in FMRP binding and that there is specificity of binding and regulation even among a long cohort of transcripts. A bias towards long transcripts in our CLIP score calculations could arise due to 3' or 5' sequencing bias in our TRAP data and subsequent underestimation of the abundance of long transcripts. To rule out this possibility, we recalculated CLIP scores

using TRAP reads mapping only within a 1 kb window at either the beginning or end of each transcript to determine transcript abundance and normalize our CLIP data. CLIP scores and the length of the predicted FMRP targets were unaffected by the transcript regions used (*Figure 3—figure supplement 1C,D*). Taken together these findings indicate that our methods can successfully discriminate FMRP targets and non-targets and the length effects seen are unlikely to be the result of length bias in our sequencing data. We also confirmed that our identification of FMRP-regulated transcripts was not driven by transcript abundance by comparing FMRP targets to a random set of expression-matched transcripts with either low or no FMRP binding and again observed a decrease in *Fmr1* KO TRAP only of the bound transcripts (*Figure 3—figure supplement 1E*).

## FMRP regulated mRNAs are enriched in specific biological functions

To further characterize the FMRP regulated transcriptome in CA1 neurons, we analyzed the biological function of the mRNAs bound by FMRP and down-regulated in *Fmr1* KO mice (*Figure 4A*). Gene ontology (GO) terms associated with FMRP binding included many neuron-specific functions which are directly related to known Fragile X phenotypes including synaptic plasticity, social behaviors, learning and memory, dendritic spine development and Ras signaling. Regulation of transcription was also a significantly associated GO term which is consistent with previous findings from our lab that epigenetic modifications causing behavioral deficits are present in the *Fmr1* KO mouse (*Korb et al., 2017*).

As expected, given our findings that FMRP targets are down-regulated in CA1 neurons in the *Fmr1* KO mouse, genes down-regulated in our TRAP data were enriched for many of the same GO terms (*Figure 4A*). The one notable exception was transcripts involved in axonal transport, microtubule based movement and microtubule organization that were enriched for FMRP binding but were not differentially expressed by TRAP. FMRP has been shown to be present in axonal granules (*Akins et al., 2017*; *Akins et al., 2012*; *Christie et al., 2009*) and may be directly involved in axonal transport (*Wang et al., 2015*) or regulation of axonal protein expression (*Akins et al., 2017*). Our findings suggest that mRNAs encoding proteins involved in axonal transport are bound by FMRP but that this binding may have an alternative functional outcome that does involve changes in mRNA abundance and/or ribosome association of the transcripts and thus is not detectable by TRAP.

While most transcripts were down-regulated in CA1 neurons in the absence of FMRP, a minority were up-regulated, and we examined their biological functions. mRNAs up-regulated in CA1 neurons in *Fmr1* KO mice were associated with GO terms related to protein synthesis and included a large number of mRNAs encoding ribosomal proteins (*Figure 4—figure supplement 1A*). These transcripts are not direct targets of FMRP but their up-regulation may be due to increased activation of an upstream pathway such as mTOR (*Hoeffer et al., 2012*; *Sharma et al., 2010*). Components of the proteasome were also up-regulated in the *Fmr1* KO possibly as a compensatory mechanism to maintain protein homeostasis in a system with excessive protein synthesis. mRNAs encoding proteins involved in mitochondrial function were also up-regulated suggesting possible mitochondrial dysfunction in the *Fmr1* KO which could impact neuronal activity. Mitochondrial dysfunction was recently reported in dfmr1 mutant flies (*Weisz et al., 2018*) and in adult neuronal stem cells from *Fmr1* KO mice (*Liu et al., 2018*) but it remains to be tested whether this phenotype is present in mature neurons or contributes to Fragile X phenotypes.

To more broadly assess the biological function of FMRP in CA1 using a diverse set of literature-curated gene sets, we performed Gene set enrichment analysis (GSEA) on our CLIP score ranked list and *Fmr1* KO TRAP fold change ranked list (*Figure 4B* and *Figure 4—figure supplement 1B*). This is a computational method that determines whether an a priori defined set of genes shows statistically significant, concordant differences between two biological states (e.g. bound vs unbound by CLIP or down-regulated vs up-regulated by TRAP) (*Mootha et al., 2003*; *Subramanian et al., 2005*). The algorithm takes a user supplied ranked list of genes and performs an unbiased search for gene sets that are statistically enriched at either end of the ranked list. Strikingly, gene sets related to circadian gene expression were significantly enriched among FMRP targets and were down-regulated in *Fmr1* KO. Other notable findings revealed by this analysis were a decrease in expression of genes known to be regulated by H3K27 methylation and the polycomb repressor complex PRC2 that methylates this histone residue, and FMRP direct or indirect regulation of a number of signaling pathways (*Figure 4B*), many of which have previously been shown to be dysregulated and linked to Fragile X

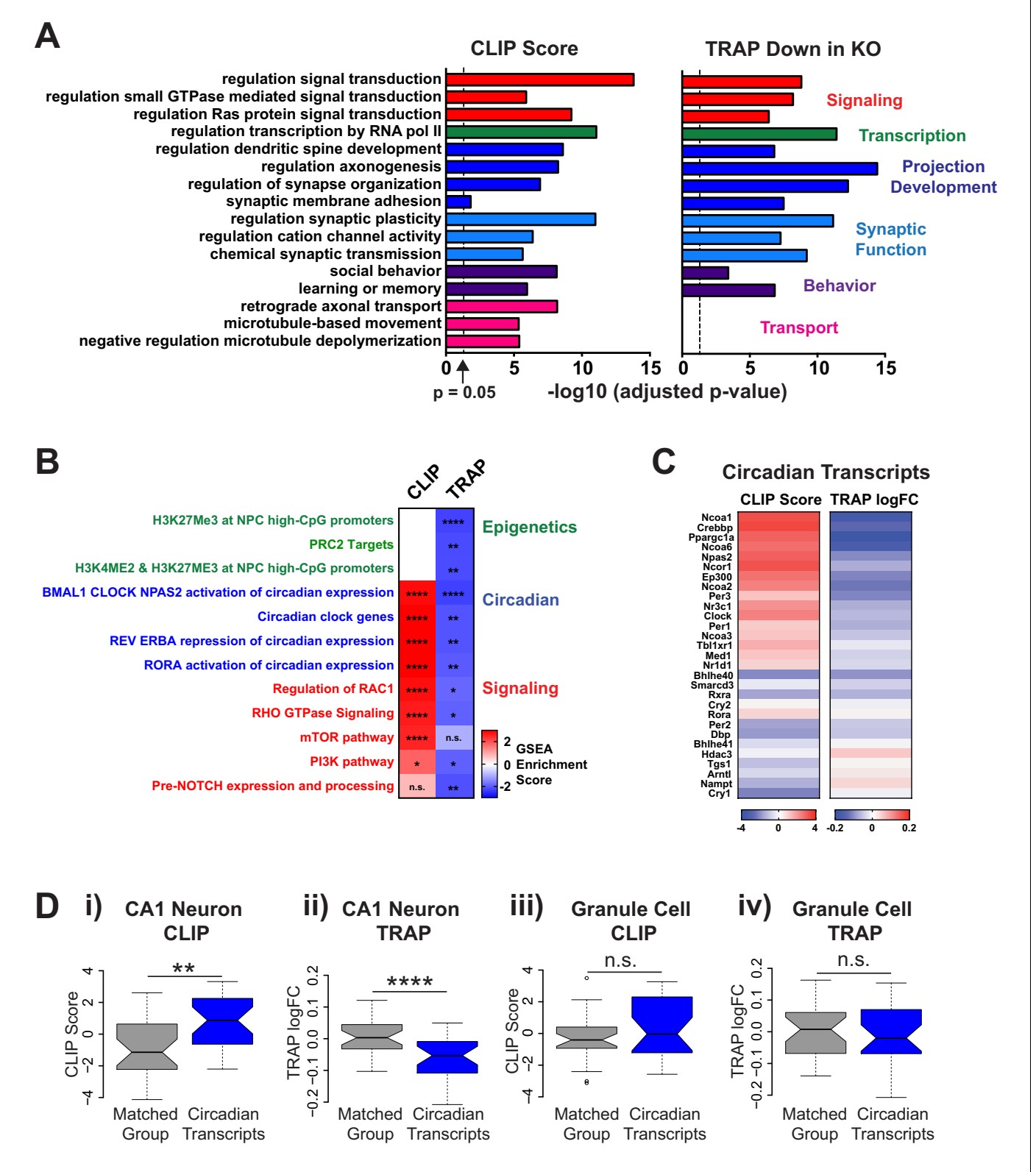

**Figure 4.** FMRP targets genes involved in neuronal function and circadian rhythm in CA1 neurons. (A) Selected results from gene ontology analysis of CA1 transcriptome ranked by FMRP binding (CLIP Score) or negative log fold change *Fmr1* KO vs WT TRAP. (B) Selected results from gene set enrichment analysis (GSEA) of the same ranked data sets. Heatmap represents normalized enrichment score (NES) with FDR values indicated (* FDR < 0.05; ** FDR < 0.01, *** FDR < 0.001, **** FDR < 0.0001, n.s. not significant). The NES reflects the degree to which a gene set is overrepresented

*Figure 4 continued on next page*

Figure 4 continued

at the top or bottom of a ranked list of genes normalized for differences in gene set sizes. (C) Heatmaps of CLIP score and log$_2$ fold change *Fmr1* KO vs WT TRAP for transcripts in the 'BMAL1 CLOCK NPAS2 activation of circadian expression' gene set from the Reactome database (*Fabregat et al., 2018*). (D) Circadian genes encode transcripts that are specifically bound and regulated by FMRP in CA1 neurons. Distribution of CLIP score (i, iii) and log$_2$ fold change *Fmr1* KO vs WT TRAP (ii, iv) for circadian genes in the 'BMAL1 CLOCK NPAS2 activation of circadian expression' Reactome gene set are shown relative to a set of randomly selected genes from the relevant transcriptome matched both for the number of genes represented and transcript length (Matched Group). Data are show for CA1 neurons (i, ii) and cerebellar granule cells (iii, iv). P-values were calculated using the Wilcoxon rank sum test (**p<0.01, ****p<0.0001, n.s. not significant).

The online version of this article includes the following figure supplement(s) for figure 4:

**Figure supplement 1.** Genes related to protein synthesis and metabolism and mitochondrial function are up-regulated and circadian genes are down-regulated in *Fmr1* KO CA1 neurons.

**Figure supplement 2.** Circadian-relevant FMRP targets are down-regulated by quantitative PCR in *Fmr1* KO mouse hippocampus.

**Figure supplement 3.** Loss of FMRP disrupts CA1 circadian oscillations in gene expression.

---

phenotypes including RAC1, mTOR and PI3K (*Gross et al., 2015a*; *Gross et al., 2015b*; *Pyronneau et al., 2017*; *Sharma et al., 2010*).

## FMRP regulation of circadian rhythm mRNAs

We chose to focus on the intriguing observation that FMRP regulates circadian transcripts in CA1 hippocampal neurons. Whilst a role for clock genes and circadian oscillations in gene expression is well established within the master clock in the suprachiasmatic nucleus (SCN) of the hypothalamus, the importance of circadian rhythms in peripheral brain regions has only more recently been recognized. The FMRP CLIP score and log fold change in the *Fmr1* KO are shown for key circadian genes involved in establishing circadian patterns of gene expression (*Figure 4C*). A high degree of FMRP binding was detected for a large subset of these genes and this was associated with a widespread down-regulation across the gene set in the *Fmr1* KO. A higher CLIP score correlated with a larger fold change in the *Fmr1* KO but down-regulation of genes not directly bound by FMRP was also observed most likely due to a global disruption of the circadian pathways in the absence of FMRP.

To further validate the enrichment of binding on circadian mRNAs, we compared binding and regulation of these mRNAs to a randomly selected set of transcripts matched for length distribution (*Figure 4D*; n = 29). The circadian mRNAs had significantly higher CLIP scores (Wilcoxon rank sum test, p=0.0037) and were significantly more down-regulated in *Fmr1* KO TRAP (Wilcoxon rank sum test, p=3.75×10–5) compared to the matched group of transcripts in CA1 neurons. A decrease in the abundance of three of the top FMRP-regulated circadian transcripts (Ppargc1a, Npas2 and Ncoa2) was validated by RT-PCR in hippocampal tissue for WT and Fmr1 KO littermates (*Figure 4—figure supplement 2*; Student's t-test, Ppargc1a p=0.0001, Npas2 p=0.0213, Ncoa2 p=0.0001) and a decrease in circadian mRNAs in CA1 *Fmr1* KO TRAP was also observed in reanalysis of the Ceolin et al. TRAP dataset, providing further confirmation of these findings (*Figure 4—figure supplement 1C*; Wilcoxon rank sum test p=0.0087).

In contrast, no significant difference was found between the circadian mRNAs and a matched set of transcripts in granule cell CLIP or TRAP data (*Figure 4D* iii,iv; Wilcoxon rank sum test, p=0.224 and p=0.865 respectively) despite similar expression of these genes across the two cell types (*Figure 4—figure supplement 1D*), suggesting that FMRP may regulate circadian rhythm in a cell-type-dependent manner.

The data shown so far indicate altered expression of circadian genes in CA1 in the absence of FMRP, however they do not address whether circadian oscillations in gene expression are altered, since all TRAP data was acquired at a comparable time of day (approximately 10am-12pm). We examined four circadian transcripts in CA1 tissue from Fmr1 KO mice and their WT littermates that have been shown to have the greatest oscillations in human hippocampus across the circadian cycle (*Li et al., 2013*). Of these, three showed a significant difference in expression in the Fmr1 KO at at least one time point while having comparable expression at other times during the circadian cycle (*Figure 4—figure supplement 3*). Two control transcripts without circadian oscillation are also shown. Pkp4, is an FMRP target that shows consistent down regulation across the circadian cycle and the other, Pou3f1, is a CA1-specific transcript that is not regulated by FMRP (*Figure 4—figure supplement 3E–F*). These data suggest a possible dysregulation of circadian oscillations of clock

gene expression in CA1 in the absence of FMRP and motivated us to look for circadian effects on hippocampus-dependent behaviors in Fragile X.

## Memory deficits in *Fmr1* KO mice are dependent on time of day

Both CA1 neurons and clock genes have been implicated in learning and memory. Given FMRP direct actions on CA1 circadian transcripts, we explored hippocampal spatial memory as a function of circadian rhythm. To examine whether *Fmr1* KO mice exhibit circadian impairments in learning and memory, we first assessed their performance in an object location memory task during the light and dark phases of the circadian cycle. Object location memory requires the hippocampus for encoding, consolidation and retrieval and is particularly sensitive to manipulations in the dorsal CA1 (*Assini et al., 2009*; *Barrett et al., 2011*) making it an ideal task in which to assess CA1-dependent memory function across the circadian cycle. Moreover deficits were observed in this task following deletion of Bmal1 in forebrain circuits indicating a direct role for the circadian clock in this learning behaviour (*Snider et al., 2016*). BMAL1 (also known as ARNTL) is a transcription factor which as a heterodimer with CLOCK regulates transcription of circadian genes and plays a pivotal role in the mammalian feedback loop responsible for generating molecular circadian rhythms.

The object location memory task is based on a rodent's innate preference for novelty and tests the animal's ability to remember the location of the objects presented during the initial training session (*Figure 5A*). During the training session, the mouse explores two identical objects placed at the northwest and northeast corners of a Plexiglas arena. One hour later the mouse is returned to the same arena in which one of the two objects has been moved. If the mouse recalls the original position of the objects, it should preferentially explore the moved object. In order to examine the effect of time-of-day on performance in the novel object location task, we used a balanced crossover design such that half of the cohort was tested first at ZT2 (day) and then 4 days later at ZT14 (night) and the other half first at ZT14 and then at ZT2. A different set of objects was used for the two tests and the sets of objects were randomized between genotype and time-of-day. All testing was performed under dim red light to prevent disruption to the circadian cycle by nocturnal light exposure and to ensure parallel testing conditions across both times of day.

Preference for the displaced object was determined by calculating a discrimination index [(time spent with displaced object – time spent with unmoved object)/(time spent with displaced object + time spent with unmoved object)]. A significant positive value indicates a preference for the displaced object and that the mouse remembers the original object positions. Consistent with intact spatial memory, WT mice showed a clear preference for the displaced object at both times of day (*Figure 5Bi*; One sample t-test vs theoretical 0 mean: day p=0.0004, night p=0.0012, n = 10 mice). In contrast the KO mice showed a deficit in spatial memory, and this was only evident during the day (*Figure 5Bi*; One sample t-test vs theoretical 0 mean: day p=0.57, night p=0.0015, n = 10 mice). This was not due to a difference in locomotor activity or exploration during the day and night as the KO mice showed similar total exploration time during the testing phase for the two objects at both times of day (*Figure 5Bii*; two-way ANOVA effect of time of day F(1,36) = 0.00201, p=0.965, n = 10 mice per genotype). Interestingly, we observed an increase in object exploration in the KO mice compared to their WT littermates (*Figure 5Bii*; two-way ANOVA effect of genotype F(1,36) = 9.729, p=0.0036, n = 10 mice per genotype), which may be linked to the hyperactivity phenotype previously reported in these animals, but this behavioral difference between the genotypes was not dependent on time-of-day indicating that only selective behavioral deficits in the Fmr1 KO mice are likely to show aberrant circadian regulation.

We further tested circadian dependent memory in a second learning paradigm, contextual fear conditioning, which involves both the hippocampus and amygdala. This test involves placing the animal in a novel environment and providing an aversive stimulus in the form of a foot shock (*Figure 5C*). When the animal is returned to the same environment, if it remembers a formed association between the context and the aversive stimulus, it will demonstrate a fear-related freezing response. To examine the effect of time of training and time of recall, we trained the mice at either ZT2 or ZT14 and then tested their recall by placing them in the same context 24, 36 and 48 hr after the initial training session. Consistent with most previous reports (*Dobkin et al., 2000*; *Peier et al., 2000*; *Van Dam et al., 2000*), we found no deficit in contextual fear conditioning in the *Fmr1* KO when training occurred during the day (*Figure 5Di*; post hoc Bonferroni's multiple comparison test, p=0.832). However, there was a significant memory deficit in *Fmr1* KO mice trained at night

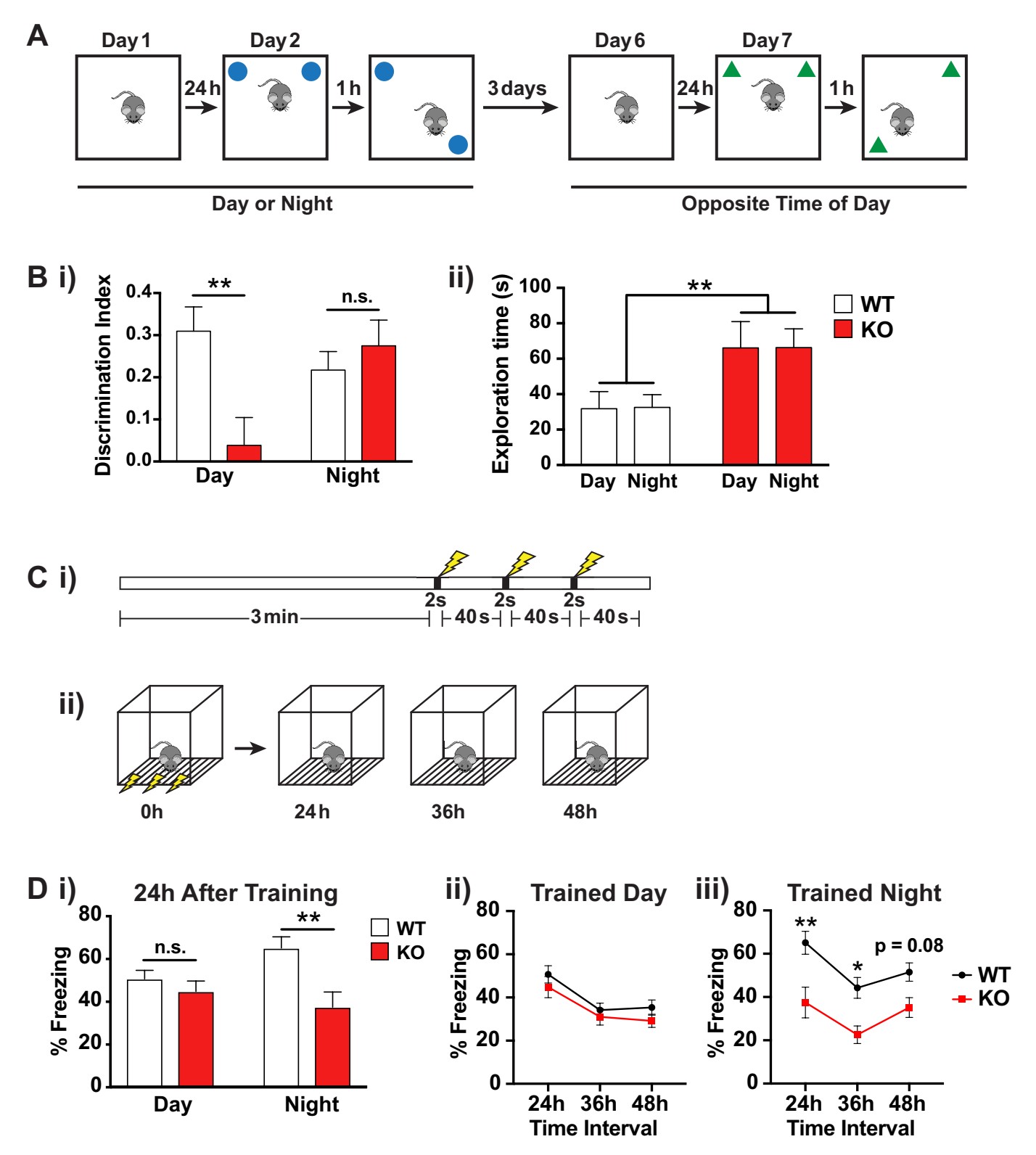

**Figure 5.** *Fmr1* KO mice memory deficits are dependent on time of day. (**A**) Schematic of spatial object recognition task. All mice were tested once during the day (9am – 11am) and once during the night (9pm – 11pm). Half of the cohort were tested first during the day and then at night and the other half were tested first at night and then during the day. On day one mice were habituated to the testing arena and allowed to explore for 10 min before being returned to their home cage. On day 2 the mice were presented with two identical objects placed at the NE and NW corners of the arena

*Figure 5 continued on next page*

*Figure 5 continued*

and allowed to explore for 10 min (sample phase). After a delay of 1 hr the mice were returned to the arena which contained the same two objects but where one of the objects had been moved to the opposite corner of the arena and allowed to explore for 10 min (choice phase). The amount of time the mice spent with the displaced object relative to the unmoved object was scored. This process was repeated after a delay of 3 days at the opposite time of day using a different set of objects. (B) Results of spatial object recognition task. (i) Discrimination index was calculated as (time spent with displaced object – time spent with unmoved object)/ (time spent with displaced object + time spent with unmoved object). A significant positive value indicates a preference for the displaced objects and an effective memory of the original object positions. A specific deficit in spatial memory was observed for *Fmr1* KO mice during the day but not at night (One sample t-test vs theoretical 0 mean: WT day p=0.0004, KO day p=0.57, WT night p=0.0012, KO night p=0.0015). Two-way ANOVA showed significant interaction between genotype and time of day ($F_{(1,35)}$ = 7.906, p=0.008, n = 10 mice per genotype). Post hoc Bonferroni's multiple comparison test was used to compare genotypes at each time of day (**p<0.01, n.s. not significant). (ii) *Fmr1* KO mice spent significantly more time exploring the objects than their WT littermates but this effect was independent of the time of day. Two-way ANOVA (genotype x time) showed significant effect of genotype ($F_{(1,36)}$ = 9.729, p=0.0036, n = 10 mice per genotype). (C) Schematic of contextual fear conditioning task. (i) Mice were placed in a novel context and allowed to explore for 3 min before being subjected to three 2 s 0.4mA shocks at 40 s intervals. (ii) 24, 36 and 48 hr after the initial training session the mice were returned to the context and the amount of freezing over a 5 min period was recorded. Mice were returned to their home cages between sessions. (D) Results of contextual fear conditioning task. (i) *Fmr1* KO mice showed significantly reduced freezing behavior compared to WT littermates when tested 24 hr after training only when trained during the night. Two-way ANOVA (genotype x time of training) showed significant main effect of genotype ($F_{(1,40)}$ = 9.890, p=0.0031, n = 8–10 mice per genotype) and a significant interaction between genotype and time of day ($F_{(1,40)}$ = 4.150, p=0.048). Post hoc Bonferroni's multiple comparison test was used to compare genotypes at each time of day (**p<0.01, n.s. not significant). (ii,iii) Both genotypes showed a similar pattern of recall across the three testing sessions suggesting that the time of day of the training event rather than the time of day of recall was the main factor in the observed deficit. A significant effect of genotype was observed only when mice were trained during the night (Two-way ANOVA, main effect of genotype $F_{(1,18)}$ = 12.36, p=0.0025). Post hoc Bonferroni's multiple comparison test was used to compare genotypes at each time interval (*p<0.05, **p<0.01). Mean ± SEM is shown for all graphs.

(*Figure 5Di*; post hoc Bonferroni's multiple comparison test, p=0.0023). Both genotypes showed a similar pattern of recall across the three testing sessions suggesting that the time of day of the training event rather than the time of day of recall was the main factor in the observed deficit (*Figure 5Dii, iii*).

## Discussion

Understanding the transcripts which FMRP binds provides an essential lens through which to view the pathophysiology of FXS. However, given the involvement of specific neuronal populations in FXS, interrogation of its relevant target transcripts necessitates cell-type-specific approaches. Here we develop FMRP cTag CLIP, which utilizes conditional Cre-lox-dependent epitope tagging of endogenous FMRP followed by CLIP to define cell-type-specific targets. This new technology allows a complex tissue such as the brain (here hippocampus and cerebellum) to be molecularly dissected in a non-invasive manner to directly examine cell-specific protein-RNA regulatory interactions in the brain. We have previously used this approach to identify differences in alternative polyadenylation in cerebellar excitatory (granule) neurons and inhibitory Purkinje neurons (*Jereb et al., 2018*), and differences in Nova-regulated alternative splicing in both excitatory and inhibitory neurons of the hippocampus and cerebellum (*Saito et al., 2018*). Here we employ FMRP cTag CLIP to identify FMRP-bound transcripts in CA1 pyramidal neurons and to identify previously unrecognized roles for FMRP in these neurons that relate to ASD and circadian rhythm.

To facilitate normalization of FMRP binding to overall transcript abundance, we undertook here TRAP analysis of ribosome-bound transcripts in the same CA1 neurons (by driving epitope tagging of the endogenous ribosomal protein RPL22 using the same Camk2a Cre driver that we used to tag FMRP). This in turn allowed us to demarcate moderately to highly stringent sets of FMRP target transcripts in CA1 neurons (*Figure 2B*) and to pursue their biology in more detail. Interestingly, we find that FMRP targets are globally decreased in *Fmr1* KO mice in both CA1 hippocampal neurons and cerebellar granule cells suggesting that FMRP may stabilize its target transcripts, perhaps either in conjunction with or downstream of its effects on their translation. This is consistent with prior observations made in CA1 neurons using CA1 TRAP alone in combination with our previously published list of whole brain FMRP targets (*Ceolin et al., 2017*; *Thomson et al., 2017*). A decrease in abundance but not ribosome association of FMRP targets has also been reported in *Fmr1* KO adult neuronal stem cells (*Liu et al., 2018*). We have now confirmed and extended these studies by refining

this observation specifically to FMRP-bound CA1 transcripts and demonstrating that the decrease in transcript abundance upon loss of FMRP is directly proportional to the amount FMRP binding.

The exact mechanism through which loss of FMRP binding leads to a decrease in mRNA abundance remains to be determined. FMRP may stabilize its target mRNAs directly as has been suggested for PSD95 (*Zalfa et al., 2007*) or changes in ribosome association and translation of FMRP targets in the *Fmr1* KO may result in compensatory change in RNA abundance (*Liu et al., 2018*). Indeed, it is well established that mRNA translation and stability are inherently linked (*Roy and Jacobson, 2013*). Given our previous findings that FMRP is associated with stalled ribosomes on targets identified by CLIP (*Darnell et al., 2011*), we suggest that FMRP may protect these mRNAs from degradation. Thus, while stalled ribosomes may often target the associated mRNA for degradation (*Doma and Parker, 2006*; *Hayes and Sauer, 2003*; *Sunohara et al., 2004*; *Tsuboi et al., 2012*), FMRP binding may directly or indirectly block such degradation. Consistent with this model, a recent ribosome profiling study examining the cortex of Fmr1 KO mice showed that many of the same gene ontology groups are significantly changed in their ribosome association as a result of alterations in translation elongation in the same direction as seen in our data (*Das Sharma et al., 2019*).

While our studies have strengthened the link between FMRP binding to changes in mRNA abundance, future research will be needed to more clearly establish underlying mechanisms, and to demonstrate how translation and protein levels are altered in specific cell types in vivo in the Fragile X brain, using, for example, new ribosome profiling or single cell-type proteomics methods. Such studies, in combination with the data presented here, will provide a more complete understanding of the complex post-transcriptional regulation of mRNAs by FMRP and may elucidate additional functional consequences of loss of this RNA-binding protein.

Prior work has established that FMRP mRNA targets identified by whole-brain CLIP are significantly enriched for transcripts arising from protein coding loci that are genetically altered in autism and schizophrenia (*Darnell et al., 2011*; *Iossifov et al., 2012*; *Purcell et al., 2014*; *Steinberg and Webber, 2013*). Our data extend these observations, indicating that the higher the probability of a gene being causative to ASD phenotypes, the more highly its transcript is bound by FMRP in CA1 neurons. Whilst FMRP binding to autism-relevant transcripts is observed in both CA1 neurons and cerebellar granule cells, we find that CA1 targets have significantly higher ASD gene scores compared to granule cell targets (*Figure 3D*). These observations point to the possibility that FMRP plays an important role in CA1 excitatory neurons, relative to cerebellar excitatory neurons, and suggests that loss of FMRP in the CA1 is relevant to the autism-related phenotypes associated with FXS.

The overlap with ASD genes is likely driven by the fundamental observation that FMRP regulates transcripts that are biologically important in brain function and are evolutionarily constrained (exhibiting genetic loss of function intolerance) (*Iossifov et al., 2012*), and reflects the enrichment of binding we observe on long transcripts (*Figure 3E*). It is therefore worth noting that we have made every attempt to minimize and test for length bias in our data, including the use of RPKM in our CLIP score determination, the use of parallel approaches (CLIP, TRAP and FACS RNA-Seq) and validation of our results by RT-qPCR. Thus, we think it is unlikely that sequencing bias has a significant effect on our determination of FMRP targets, although it is impossible to completely rule out any such effect, which could in turn lead to some influence on our observation that FMRP targets are enriched for long transcripts. Perhaps the most compelling evidence that our selection of targets is biologically relevant and not an artifact of length bias is the fact that the differences seen in the Fmr1 KO by TRAP are specific to targets even when comparing to length-matched controls (*Figure 3F*).

Using an unbiased approach to study CA1 neuronal FMRP targets, we identified circadian genes as the most significant set of CA1 transcripts enriched in both CLIP-binding and showing changes in ribosome association in *Fmr1* KO by CA1 TRAP (*Figure 4B*). Among our list of stringent CA1 targets are Npas2, a core component of the circadian clock, Ppargc1a, a key component of the circadian oscillator that integrates the mammalian clock and energy metabolism (*Liu et al., 2007*), Ncoa2, a coactivator of the key transcription BMAL1:CLOCK regulatory complex of the circadian clock (*Stashi et al., 2014*) and Crebbp, a coactivator of circadian transcription and a regulator of circadian behavior downstream of the circadian clock (*Hung et al., 2007*; *Lim et al., 2007*; *Maurer et al., 2016*).

It has been previously noted that children with Fragile X syndrome experience sleep difficulties, including disturbances consistent with circadian clock shifts (early rising or late onset sleep)

(*Kaufmann et al., 2017*; *Kronk et al., 2010*; *Richdale, 2003*). Moreover, sleep disturbances are more prevalent among Fragile X children with ASD (*Kaufmann et al., 2017*). *Fmr1* KO mice show limited changes in their overall sleep-wake cycle behavior (*Boone et al., 2018*; *Saré et al., 2017*; *Zhang et al., 2008*) however they do exhibit abnormal sleep architecture with fewer bouts of REM sleep (*Boone et al., 2018*) and a shorter free-run period of locomotor activity in continuous darkness indicating a slightly shortened internal circadian cycle in the absence of light entrainment (*Zhang et al., 2008*). In addition, cortical neurons in *Fmr1* KO mice have abnormally high firing rates during sleep (*Gonçalves et al., 2013*) and *Fmr1* KO CA1 pyramidal cells are hyperactive in all sleep and wake states (*Boone et al., 2018*). Effects on circadian activity are more dramatic if the FMRP paralog FXR2 is also deleted. *Fmr1/Fxr2* double KO mice display arrhythmic circadian activity (*Zhang et al., 2008*) and a similar result is seen in dfmr1-null flies which have only one FMR1 family gene (*Dockendorff et al., 2002*; *Morales et al., 2002*), suggesting that in mammals the FXR2 paralog may compensate for some of the effects of loss of FMR1. Circadian rhythm-dependent alteration of gene expression has been observed in dfmr1 mutant fly heads (*Xu et al., 2012*) and liver tissue from *Fmr1* KO, *Fxr2* KO and *Fmr1/Fxr2* double KO mice (*Zhang et al., 2008*) and a decrease in expression of circadian-related genes has been observed in lymphoblastoid cell lines from FXS patients (*Mullegama et al., 2015*). However, in the suprachiasmatic nucleus, the central brain region for regulation of circadian rhythm, *Fxr2* was found to have a greater effect on circadian gene expression, with significant differences in the expression of core circadian genes observed only in *Fxr2* KO and *Fmr1/Fxr2* double KO mice but not *Fmr1* KO mice (*Zhang et al., 2008*).

Circadian rhythms in mammals are regulated by the SCN in the hypothalamus which serves as the 'master clock' for the brain and body. However, clock genes are rhythmically expressed throughout many brain regions and play a critical role in the regulation of normal brain processes. In particular, the hippocampus is a convergence point of sleep and memory. Clock genes are expressed rhythmically in the hippocampus in mice and humans (*Jilg et al., 2010*; *Li et al., 2013*; *Ma et al., 2016*) and hippocampal slices continue these oscillations in culture indicating an intrinsic circadian expression independent of the SCN (*Wang et al., 2009*). Circadian rhythms have also been shown in hippocampal long term potentiation at CA1 synapses (*Chaudhury et al., 2005*), in firing rates of CA1 place cells (*Munn and Bilkey, 2012*) and in memory acquisition and recall (*Smarr et al., 2014*; *Snider et al., 2018*). Deficits in hippocampal-dependent learning tasks have previously been observed in mice following disruption of key circadian genes including *Per1*, *Cry1/2*, *Clock* and *Arnt* (*Bmal1l*) (*Jilg et al., 2010*; *Sei et al., 2006*; *Snider and Obrietan, 2018*; *Van der Zee et al., 2008*; *Wardlaw et al., 2014*). Moreover, conditional deletion of *Bmal1* in forebrain excitatory neurons but not in the SCN, using *Camk2a-Cre*, causes deficits in Barnes maze performance and novel object location and novel object recognition tasks in the absence of any change in circadian locomotor activity (*Snider et al., 2016*; *Snider and Obrietan, 2018*). These findings support a direct role for clock gene expression outside of the SCN in hippocampal function and demonstrate that circadian rhythms in learning and memory can be independent of locomotor rhythms.

Our data indicate for the first time a direct role for FMRP in the regulation of transcripts critical for circadian function in the hippocampus and allow us to establish a molecular correlate of circadian defect in excitatory neurons in the brain. We find a specific regulation of circadian transcripts in CA1 pyramidal cells but not cerebellar granule cells, despite similar expression of these transcripts in both cell types (*Figure 4D*, *Figure 4—figure supplement 1D*). To date, since in vivo cell-type-specific identification of FMRP binding and analysis of downstream regulation has only been performed in CA1 pyramidal neurons and cerebellar granule cells, it remains unclear if dysregulation of circadian transcripts exists in other circadian-relevant cell types in the *Fmr1* KO mouse and whether this may also contribute to the behavioral phenotypes observed here. Nonetheless, the finding of cell-type specific circadian control in general, and additionally that such regulation can be FMRP-dependent, are important new observations.

Given the nature of the functions attributed to CA1 neurons, aberrant regulation of circadian transcripts in this cell type suggested the possibility of a circadian defect in memory in the absence of FMRP. While the absence of FMRP in humans (*Schmitt et al., 2019*) and in animal models (*Drozd et al., 2018*; *Kazdoba et al., 2014*; *Tian et al., 2017*; *Till et al., 2015*) is well-established to cause memory defects, our studies are able to extend these observations by demonstrating that defects in at least two modes of learning and memory are dependent on the phase of the circadian cycle. Specifically, we found that in the object location memory task, *Fmr1* KO mice had marked

deficits in spatial memory when trained and tested in the inactive phase of their circadian cycle (light phase/day) (*Figure 5B*). Whereas in contextual fear conditioning, a memory deficit was observed only when the animal was trained during the active phase of their circadian cycle (dark phase/night) (*Figure 5D*). Our findings that *Fmr1* KO mice show deficits in object location memory but not contextual fear conditioning during the day, is consistent with findings of others (*Dobkin et al., 2000*; *King and Jope, 2013*; *Peier et al., 2000*; *Van Dam et al., 2000*) but reveal an unexpected dependence of these findings on the phase of the circadian cycle in which they are tested. Moreover, consistent with our findings that circadian transcripts are specifically regulated by FMRP in the CA1, circadian differences are not observed in other behavioral characteristics that are less dependent on the hippocampus, including hyperactivity (*Figure 5B*, *Saré et al., 2016*) and anxiety (*Saré et al., 2016*). Memory consolidation, including spatial memory encoded by hippocampal place cells, is believed to occur during sleep (*Lee and Wilson, 2002*; *Ólafsdóttir et al., 2018*; *Wilson and McNaughton, 1994*) and FMRP has been shown to play an important role in sleep-dependent synaptic homeostasis in *Drosophila* (*Bushey et al., 2011*). Thus, the deficits observed in spatial and contextual memory may be the result of sleep disturbance or an independent function of the circadian genes in learning and memory within the hippocampus. While our results combined with previous findings of others are suggestive of a direct link between altered circadian gene expression in the hippocampal CA1 region and altered CA1-dependent circadian behaviors, it remains to be tested whether other cell types or brain regions also contribute to these phenotypes. Future work, for example using CA1 specific deletion of FMRP, will be necessary to establish a direct link. Together our findings indicate that altered circadian brain rhythms may contribute to the deficits in hippocampal-dependent learning and memory in FXS.

# Materials and methods

**Key resources table**

| Reagent type (species) or resource | Designation | Source or reference | Identifiers | Additional information |
|---|---|---|---|---|
| Strain (*M. musculus*), strain background (*C57BL6/J*) | B6.Cg-Tg(Camk2a-cre)T29-1Stl/J | Jackson Laboratory | RRID:IMSR_JAX:005359 | Referred to as *Camk2a-Cre.* |
| Strain (*M. musculus*), strain background (*C57BL6/J*) | B6N.129-Rpl22$^{tm1.1Psam}$/J | Jackson Laboratory | RRID:IMSR_JAX:011029 | Referred to as RiboTag. |
| Strain (*M. musculus*), strain background (*C57BL6/J*) | B6.Cg-Gt(ROSA)26Sor$^{tm14(CAG-tdTomato)Hze}$/J | Jackson Laboratory | RRID:IMSR_JAX:007914 | Referred to as TdTomato. |
| Strain (*M. musculus*), strain background (*C57BL6/J*) | B6.129P2-Fmr1$^{tm1Cgr}$/J | Gift from W.T. Greenough. | RRID:IMSR_JAX:003025 | Referred to as *Fmr1* KO. |
| Strain (*M. musculus*), strain background (*C57BL6/J*) | Neurod1-cre (RZ24-CRE) | Gift from M.E. Hatten. | | |
| Strain (*M. musculus*), strain background (*C57BL6/J*) | Fmr1-cTag | *Van Driesche et al., 2019* | | |
| Antibody | NeuN, guinea pig polyclonal | Millipore | Millipore Cat# ABN90P, RRID:AB_2341095 | For IF (1:2000) |
| Antibody | GFP polyclonal antibody, rabbit | Invitrogen | Molecular Probes Cat# A-11122, RRID:AB_221569 | For IF (1:2000) |
| Antibody | Anti-FMRP antibody, rabbit polyclonal | Abcam | Abcam Cat# ab17722, RRID:AB_2278530 | For IF (1:200) |
| Antibody | Anti-HA tag, rabbit polyclonal | Abcam | Abcam Cat# ab9110, RRID:AB_307019 | For IP (20–80 µg/ml depending on Cre driver) |

*Continued on next page*

*Continued*

| Reagent type (species) or resource | Designation | Source or reference | Identifiers | Additional information |
|---|---|---|---|---|
| Antibody | anti-GFP antibodies HtzGFP19C8 and HtzGFP19F7, mouse monoclonal | PMID: 19013281 | Heintz Lab; Rockefeller University Cat# Htz-GFP-19C8, RRID:AB_2716737 Heintz Lab; Rockefeller University Cat# Htz-GFP-19F7, RRID:AB_2716736 | For IP (25 µg each antibody for 1.2 ml lysate prepared from 8 to 10 animals) |
| Antibody | Anti-BrdU, mouse monoclonal [IIB5] | Abcam | Abcam Cat# ab8955, RRID:AB_306886 | For IP (5 µg per pooled RT reaction) |
| Antibody | anti-p0071, guinea pig polyclonal | Progen | Progen Cat# GP71 | p0071 is also known as Pkp4. For WB (1:1000) |
| Antibody | Anti-ASCIZ, rabbit polyclonal | Millipore | Millipore Cat# AB3271, RRID:AB_11215293 | ASCIZ is also known as Atmin. For WB (1:5000) |
| Antibody | p190-A RhoGAP, rabbit polyclonal | Cell Signaling | Cell Signaling Technology Cat# 2513, RRID:AB_2232820 | p190-A RhoGAP is also known as Arhgap35. For WB (1:1000) |
| Antibody | Anti-HA tag, goat polyclonal | Abcam | Abcam Cat# ab9134, RRID:AB_307035 or Abcam Cat# ab215069, RRID:AB_2811264 | For WB (1:10,000) |
| Antibody | Anti-Ribosomal P, human polyclonal | US Biological | US Biological Cat# R2031-25A, RRID:AB_2146244 | For WB (1:10,000) |
| Software, algorithm | EthoVision tracking software | Noldus Information Technology | RRID:SCR_000441 | |
| Software, algorithm | Behavioral Observation Research Interactive Software (BORIS) | doi: 10.1111/2041-210X.12584 | | |
| Software, algorithm | FreezeFrame three software | Coulbourn Instruments | RRID:SCR_014429 | |
| Software, algorithm | STAR | PMID: 23104886 | RRID:SCR_015899 | |
| Software, algorithm | featureCounts | PMID: 24227677 | RRID:SCR_012919 | |
| Software, algorithm | DESeq2, Bioconductor | PMID: 25516281 | RRID:SCR_015687 | |
| Software, algorithm | CLIP Tool Kit (CTK) | PMID: 27797762 | | |
| Software, algorithm | RSeQC | PMID: 22743226 | RRID:SCR_005275 | |
| Software, algorithm | GenomicRanges, Bioconductor | PMID: 23950696 | RRID:SCR_000025 | |
| Software, algorithm | Limma, Bioconductor | PMID: 25605792 | RRID:SCR_010943 | |
| Software, algorithm | GOrilla | PMID: 19192299 | RRID:SCR_006848 | |
| Software, algorithm | GSEA | PMID: 16199517 | RRID:SCR_003199 | |
| Commercial assay or kit | Quant-iT RiboGreen RNA Assay Kit | ThermoFisher Scientific | Cat# R11490 | |
| Commercial assay or kit | High Pure RNA isolation kit | Roche | Cat# 11828665001 | |
| Commercial assay or kit | Ribo-Zero rRNA Removal Kit (Human/Mouse/Rat) | Illumina | Cat# MRZH11124 | |
| Commercial assay or kit | Dynabeads mRNA Purification Kit | ThermoFisher Scientific | Cat# 61006 | |

*Continued on next page*

Continued

| Reagent type (species) or resource | Designation | Source or reference | Identifiers | Additional information |
|---|---|---|---|---|
| Commercial assay or kit | TruSeq RNA library prep kit | Illumina | Cat# RS-122–2001 | |
| Commercial assay or kit | iScript cDNA Synthesis Kit | Bio-Rad | Cat# 1708891 | |
| Commercial assay or kit | FastStart SYBR Green Master | Roche | Cat# 04673484001 | |

## Mice

Animals were maintained in a temperature- and light-controlled environment with a 12/12 hr light/dark cycle (lights on at seven am) and were treated in accordance with the principles and procedures of the National Institutes of Health Guide for the Care and Use of Laboratory Animals. All mouse experiments were approved by The Rockefeller University Institutional Animal Care and Use Committee. Unless stated otherwise, animals used for these studies were male littermates anesthetized with isoflurane and euthanized by decapitation at 28–32 days postnatal. All procedures during the dark cycle were performed under red light.

All mouse lines used in these studies were on the C57BL/6 genetic background. B6.Cg-Tg (Camk2a-cre)T29-1Stl/J (*Camk2a-Cre*), B6N.129-Rpl22$^{tm1.1Psam}$/J (RiboTag) and B6.Cg-Gt(ROSA) 26Sortm14(CAG-tdTomato)Hze/J (tdTomato) were purchased from Jackson Laboratories. B6.129P2-Fmr1$^{tm1Cgr}$/J (*Fmr1* KO) mice were a generous gift from W.T. Greenough maintained for multiple generations in our own facilities. Neurod1-cre mice were a generous gift from M.E. Hatten, Rockefeller University, and were generated as part of the Gene Expression Nervous System Atlas Project (GENSAT) as previously described (*Gong et al., 2003*). The *Fmr1*-cTag mouse has been described and characterized elsewhere (*Van Driesche et al., 2019*). Briefly this mouse line was generated by introducing loxP sites either side of the terminal exon of the *Fmr1* gene followed by a downstream AcGFP-tagged version of the terminal exon and surrounding intronic sequences. Thus, either FMRP or AcGFP-tagged FMRP can be expressed from the cTag allele in a mutually exclusive manner, dependent on Cre expression.

In order to obtain WT and *Fmr1* KO male littermates for our studies all *Fmr1* breeding schemes used *Fmr1* + /- females bred to *Fmr1* +/Y males. Mice for CLIP experiments were generated by breeding male Cre mice with homozygous *Fmr1*-cTag females to give Cre +/-; Fmr1-cTag +/Y male offspring. Mice for TRAP experiments were generated by breeding male Cre mice with Fmr1 +/-; RiboTag +/+ females to give Cre +/-; RiboTag +/-; Fmr1 KO and Cre +/-; RiboTag +/-; Fmr1 WT littermates. Mice for FACS RNA-Seq experiments were generated by breeding male *Camk2a-Cre* mice with tdTomato +/+ females to give *Camk2a-Cre* +/-; tdTomato + /- offspring.

## Immunofluorescence

Mice were anesthetized with isoflurane and transcardially perfused with PBS containing 10 U/ml heparin followed by perfusion with ice-cold PBS containing 4% paraformaldehyde. After perfusion, animals were decapitated, and intact brains removed and postfixed overnight in 4% paraformaldehyde in PBS at 4°C. Brains were then transferred to PBS with 15% sucrose for 24 hr followed by PBS with 30% sucrose for a further 24 hr and then embedded and frozen in OCT medium. 30 µm coronal sections were cut using a Leica CM3050 S cryostat and stored in a cryoprotectant storage solution containing 30% ethylene glycol, 25% glycerol and 45% PBS at −20°C until use.

Immunofluorescence was performed in free floating sections. Sections were wash in PBS and then incubated with 1% SDS in PBS for 5 min at room temperature for antigen retrieval. Sections were permeabilized by incubation in 0.3% Triton-X in PBS for 15 min and then incubated in blocking buffer (10% normal horse serum, 0.1% Triton in PBS) for 1 hr. Primary antibody incubations were performed overnight at 4°C [1:2000 NeuN polyclonal antibody (Millipore, ABN90P), 1:2000 GFP polyclonal antibody (Invitrogen, A11122) or 1:200 FMRP antibody (Abcam, ab17722)] in antibody dilution buffer (1% BSA, 0.05% Triton-X in PBS)] and secondary antibody incubations were performed at room temperature for 1 hr (1:1000 Alexa-Fluor conjugated secondary antibodies, Invitrogen, in antibody dilution buffer). Sections were and mounted with Vectashield mounting media containing DAPI

(Vector Laboratories) and imaged using a Keyence BZ-X710 fluorescence microscope or a Zeiss LSM 880 NLO laser scanning confocal microscope.

## FMRP cTag CLIP

For each independent biological replicate, hippocampi from 8 to 10 *Fmr1*-cTag[Camk2a-Cre] and 8–10 *Fmr1*-cTag (Cre negative) male mice aged postnatal day 28 to 32 were pooled by genotype. Hippocampi were dissected into HBSS (Hanks' Balanced Salt solution) containing 0.1 mg/mL cycloheximide and crosslinked three times on ice as described previously (*Darnell et al., 2011*). Crosslinked material was collected by centrifugation and resuspended in 1.2 mL lysis buffer (1X PBS, 1% Igepal, 0.5% deoxycholate and 0.1% SDS with protease inhibitor). Material was homogenized by mechanical homogenization and frozen at −80°C to ensure full cell lysis. To prepare for immunoprecipitation, lysates were thawed and subject to DNase and RNase treatment as described previously (*Moore et al., 2014*), using 54 µl of RQ1 DNase (Promega) and RNase A (Affymetrix, 20 U/ml) at a final dilution of 1:1,666,666. Following RNase and DNase treatment, the lysate was clarified by centrifugation at 2000 x g for 10 min, followed by an addition spin of the supernatant at 20,000 x g for 10 min. The resulting supernatant was pre-cleared by the addition of 200 µl protein G dynabeads (washed in lysis buffer). The sample was rotated for 45 min at 4°C. The resulting supernatant was used for immunoprecipitation with 200 µl of Protein G Dynabeads (Invitrogen) loaded with 25 µg each of mouse monoclonal anti-GFP antibodies 19F7 and 19C8 (*Heiman et al., 2008*). Immunoprecipitation was performed for 1.5–2 hr. After which the following washes were performed: twice with lysis buffer, twice with high salt lysis buffer (5X PBS, 1% Igepal, 0.5% deoxycholate and 0.1% SDS), twice with stringent wash buffer (15 mM Tris pH 7.5, 5 mM EDTA, 2.5 mM EGTA, 1% TritonX-100, 1% NaDOC, 0.1% SDS, 120 mM NaCl, 25 mM KCl), twice with high salt wash buffer (15 mM Tris pH 7.5, 5 mM EDTA, 2.5 mM EGTA, 1% TritonX-100, 1% NaDOC, 0.1% SDS, 1M NaCl), twice with low salt wash buffer (15 mM Tris pH 7.5, 5 mM EDTA), and twice with PNK wash buffer (50 mM Tris pH 7.4, 10 mM MgCl2, 0.5% NP-40). The second of each wash was rotated for 2–3 min at room temperature. RNA tags were dephosphorylated as described previously (*Moore et al., 2014*) and subjected to overnight 3' ligation at 16°C with a pre-adenylated linker (preA-L32) (*Moore et al., 2018*) with the following ligation reaction: 2 µl of 25 µM linker, 2 µl of T4 RNA Ligase two truncated K227Q (NEB), 1X ligation buffer (supplied with ligase), 2 µl Superasin RNase inhibitor (Invitrogen), and 8 µl PEG8000 (supplied with ligase). The beads were washed and the RNA-protein complexes [32]P-labeled, and subjected to SDS-PAGE and transfer as described (*Moore et al., 2014*). Tags were collected from nitrocellulose as described (*Zarnegar et al., 2016*) with the following exceptions: Phenol:Chloroform:IAA, 25:24:1 pH 6.6 was used for extraction, and tags were precipitated with a standard sodium acetate precipitation. Cloning was performed using the BrdU-CLIP protocol as described (*Moore et al., 2018*) with a few exceptions. RT primers were generated with six nucleotide barcode index sequences to allow for up to 24 samples to be pooled together in one MiSeq run and subsequently demultiplexed (see *Supplementary file 4* for sequences). To increase the yield for low-input samples, the *Fmr1*-cTag[Camk2a-Cre] hippocampi samples were pooled (after reverse transcription) with parallel *Fmr1*-cTag[Camk2a-Cre] cortex samples. Both *Fmr1*-cTag[Camk2a-Cre] and *Fmr1*-cTag (Cre negative) samples were also included in the pool.

### TRAP

For CA1 TRAP, hippocampi were rapidly dissected from P28 - P31 RiboTag[Camk2a-Cre] Fmr1 KO and WT littermates and homogenized in 1 ml ice-cold polysome buffer containing 20 mM Hepes pH7.4, 150 mM NaCl, 5 mM MgCl$_2$, 0.5 mM DTT, 0.1 mg/ml cycloheximide supplemented with 40 U/ml RNasin Plus (Promega) and cOmplete Mini EDTA-free Protease Inhibitor (Roche). 0.1 volumes of 10% NP-40 was added and the samples incubated for 10 min on ice. Insoluble material was removed by centrifugation at 2000 xg, 10 min, 4°C followed by further centrifugation of the supernatant at 20,000 xg, 10 min, 4°C.

100 µl lysate was mixed with 300 µl Trizol LS (Invitrogen) for RNA extraction as the input sample. The remaining lysates was precleared by incubating with 50 µl Protein A Dynabeads (Invitrogen) for 45 min at 4°C with rotation then incubated with 20 µg anti-HA (ab9110, abcam) for 2 hr at 4°C with rotation. The antibody-polysome complexes were immunoprecipitated by addition of 150 µl Protein A Dynabeads and incubation for a further 1 hr at 4°C with rotation.

Protein A Dynabeads were washed 3 × 2 min with polysome buffer containing 1% NP-40, then 4 × 2 min with high salt buffer containing 50 mM Tris pH 7.5, 500 mM KCl, 12 mM MgCl$_2$, 1% NP-40, 1 mM DTT and 0.1 mg/ml cycloheximide. Polysomes were eluted from the beads by incubation with 500 µl Trizol (Invitrogen) for 5 min at room temperature with occasion vortexing. RNA was extracted from Trizol reagent as per the manufacturer's instructions. RNA was quantified using the Quant-iT RiboGreen RNA Assay Kit (Invitrogen).

For sequencing library preparation, RNA was further purified using the High Pure RNA isolation kit with on-column DNase treatment (Roche). Ribosomal RNA was removed using the Ribo-Zero rRNA Removal Kit (Illumina) and the remaining RNA cloned using the TruSeq RNA library prep kit (Illumina). Libraries were sequenced on an Illumina HiSeq 2500 with 100 bp paired-end reads.

For cerebellar granule cell TRAP, cerebellum from 6 to 8 week old RiboTag[Neurod1-Cre] Fmr1 KO and WT littermates were dissected, homogenized and processed as for CA1 TRAP except for the following change in volumes to allow for higher RPL22-HA expression in these lysates: 500 µl lysate, 40 µg anti-HA and 300 µl Protein A Dynabeads were used per immunoprecipitation.

## Western blotting

Hippocampal tissue was homogenized in lysis buffer (25 mM Tris pH7.4, 150 mM NaCl, 1 mM EDTA pH8.0, 0.1% SDS, 0.5% sodium deoxycholate, 1% Triton X-100 supplemented with cOmplete Mini EDTA-free Protease Inhibitor (Roche). The lysates were sonicated and DNase treated with RQ1 Dnase I (Roche). Insoluble material was removed by centrifugation. Protein concentration was determined by means of a BCA protein assay kit (Pierce). Aliquots (30–80 µg) were run on NuPAGE gels (Invitrogen) and transferred to a PVDF membrane. Total protein on the membrane was stained with REVERT total protein stain (LI-COR), imaged using an Odyssey CLx imaging system (LI-COR) and then destained according to the manufacturer's protocol. Proteins of interest were probed with the appropriate antibody (Pkp4 (p0071) Progen, GP71; Atmin (ASCIZ) Millipore, AB3271; Arhgap35 (p190A RhoGAP), Cell Signaling 2513; HA-tag Abcam, ab9134 or ab215069; Ribosomal P proteins, US Biological R2031-25) and detected using an appropriate IRDye conjugated secondary antibody (LI-COR). All antibodies were diluted in Odyssey Blocking Buffer (PBS). Blots were imaged and quantified using an Odyssey CLx imaging system and Image Studio software (LI-COR). Protein bands were normalized to total protein of a similar molecular weight, using the total protein stain and a region of the lane encompassing the molecular weight of the protein of interest.

## RNA extraction and quantitative PCR

Hippocampal tissue was dissected in ice-cold HHBSS (Hanks' Balanced Salt solution buffered with 10 mM Hepes pH7.3). For CA1 micro-dissection, 400 µm slices were cut using a McIlwain tissue chopper. The slices were placed in ice-cold HHBSS, separated and the CA1 region dissected from each slice. All tissue was snap-frozen in liquid nitrogen. RNA was extracted from tissue using Trizol reagent or from homogenized tissue lysates with Trizol LS reagent according to the manufacturer's instructions. Reverse transcription was performed using iscript (Bio-Rad) and quantitative PCR using FastStart SYBR Green Master (Roche). Data was normalized to a combination of three housekeeping genes, Gapdh, Hprt and Actb.

## qPCR primer sequences

Arhgap35 fwd GTGACTCCAGAGAAACCGATAC
Arhgap35 rev GTAGATGCCTTCAGTGCTTAGT
Atmin fwd ATGCACCTCGTCAAGAGCC
Atmin rev ACATCCTTTGATTGGACAACAGT
Pkp4 fwd GAACCTGTCATACCGGCTGG
Pkp4 rev TTCCGAGTCTTTGCTGGGAGA
Arntl fwd TGACCCTCATGGAAGGTTAGAA
Arntl rev GGACATTGCATTGCATGTTGG
Ncoa2 fwd GACAGCGGCCAAATTACACC
Ncoa2 rev ATAAGCGGCTGGCGATTCTG
Npas2 fwd AAGGATAGAGCAAAGAGAGCCT
Npas2 rev CATTTTCCGAGTGTTACCAGGG
Nr1d1 fwd TTTTTCGCCGGAGCATCCAA

Nr1d1 rev ATCTCGGCAAGCATCCGTTG
Per2 fwd GAAAGCTGTCACCACCATAGAA
Per2 rev AACTCGCACTTCCTTTTCAGG
Per3 fwd AACACGAAGACCGAAACAGAAT
Per3 rev CTCGGCTGGGAAATACTTTTTCA
Pou3f1 fwd TCGAGGTGGGTGTCAAAGG
Pou3f1 rev GGCGCATAAACGTCGTCCA
Ppargc1a fwd AAGTGGTGTAGCGACCAATCG
Ppargc1a rev AATGAGGGCAATCCGTCTTCA
Gapdh fwd TGAACGGGAAGCTCACTGGCAT
Gapdh rev TCAGATGCCTGCTTCACCACCT
Actb fwd CGCCACCAGTTCGCCATGGA
Actb rev TACAGCCCGGGGAGCATCGT
Hprt fwd TCCTCCTCAGACCGCTTTT
Hprt rev CCTGGTTCATCATCGCTAATC

## Fluorescence-activated cell sorting and RNA sequencing (FACS RNA-Seq)

Hippocampi were dissected from two to four *Camk2a-Cre* +/-; tdTomato + /- male mice per biological replicate at 28–32 days postnatal. The CA1 pyramidal cells from these mice are labeled with tdTomato to enable their separation based on fluorescence. The hippocampi were washed with ice-cold HABG [Hibernate A (Brainbits) supplemented with 2% B-27 supplement (Gibco) and 0.25% Glutamax (Gibco)] and cut into 300 µm slices with a McIlwain tissue chopper. Tissue slices were washed once with papain buffer [Hibernate A minus calcium (Brainbits) supplemented with 0.25% Glutmax] and then digested with 2 mg/ml papain (Worthington Biochemical Corporation) and 50 µg/ml DNase I dissolved in papain buffer at 37°C for 30 min with mixing. Tissue was washed twice with HABG and then gently triturated ten times with a P1000 pipette tip. Triturated tissue was filtered through a cell strainer and cells pelleted by centrifugation at 1000 rpm, 5 min, 4°C. The cell pellet was resuspended in ice-cold FACS buffer (3% BSA in PBS with 100 ng/ml DAPI) and transferred to a 5 mL round Bottom polystyrene tube via a 35 µm cell strainer. Live, tdTomato positive cells were separated on a BD FACSAria Cell Sorter and collected into 3% BSA in PBS. Cells were pelleted by centrifugation and lysed in Trizol.

RNA was extracted from Trizol as per the manufacturer's protocol. The RNA was then treated with DNAse I (RNase-Free DNase Set, Qiagen) and further purified using the RNeasy minelute cleanup kit (Qiagen) according to the manufacturer's instructions. Sequencing libraries were prepared using Poly(A) RNA selection (Dynabeads mRNA Purification Kit, ThermoFisher Scientific) and the TruSeq RNA Library Prep Kit (Illumina). Libraries were sequenced on an Illumina NextSeq 500 with 150 bp paired-end reads.

## Behavioral testing

All behavior testing was conducted under red light regardless of the time of testing to ensure equivalent testing conditions for all tasks. During the dark cycle, mice were transported to the behavior rooms in black boxes to avoid disruption of the circadian clock by exposure to white light. Behavior testing was conducted either starting at 9am (ZT2) or 9pm (ZT14). Mice for novel object location were 8–12 weeks old. Mice for fear conditioning were 2–4 months old.

For novel object location, the task was conducted in a clear Plexiglas open arena (58 × 58 × 46 cm) using two types of objects (pyramids constructed of transparent plastic with colored edges and half-inch threaded gate valves). The task consisted of three phases: habituation, sample, and choice, conducted successively. During habituation, the mouse was placed in the empty arena and allowed to freely explore for 10 min. The sample phase was conducted 24 hr later. Two identical objects were fixed to the floor in the northeast and northwest corners of the box (10 cm from the walls) and the mice were allowed to explore for 10 min. The choice phase was conducted 1 hr after the sample phase in which one of the objects was moved to the opposite corner of the arena (either the NE object moved to the SE corner or the NW object moved to the SW corner, whichever object was explored less during the sample phase). The type of object used was counterbalanced across

groups. The mice were allowed to explore the objects again for 10 min. The arena and objects were cleaned with 30% (vol/vol) alcohol between each animal.

After 3 days rest, the mice went through a second round of the same 3 phases of habituation, sample, and choice but at the opposite time of day. Half of the mice were tested in the first round at ZT2 and then subsequently at ZT14 (in the second round). The other half were tested first at ZT14 and then subsequently at ZT2. For the second testing session the arena was the same but different objects were used.

EthoVision tracking software (Noldus Information Technology, Leesburg, Virginia) was used during the task. Object exploration was scored manually from the videos using Behavioral Observation Research Interactive Software (BORIS) (*Friard and Gamba, 2016*). Object exploration was scored only when the mouse's nose touched the object. All scoring was performed blind to genotype and time of testing. Mice were excluded if they spent less than a total of 5 s exploring the objects during the choice phase.

For fear conditioning, the mouse was placed in a 18 cm x 18 cm x 30 cm testing chamber where the floor consisted of metal bars through which a foot shock could be delivered (Coulbourn Instruments). The testing room was illuminated with red light and the testing chambers themselves were enclosed with no external light source. Video was recorded using infrared light only. During the training session the mice were allowed to explore the chamber for 3 min, then subjected to three 2 s 0.4 mA foot shocks at 40 s intervals. 24, 36 and 48 hr after the initial training session the mice were returned to the testing chamber and their activity recorded for 5 min before being returned to their home cage. Automated scoring of freezing behavior was performed using FreezeFrame three software (Coulbourn Instruments).

## Bioinformatics

### TRAP

Paired-end reads from input and TRAP samples were aligned to the mouse genome (GRCm38) using STAR (*Dobin et al., 2013*). Mapped reads per gene were counted using featureCounts (*Liao et al., 2014*) using GENCODE vM15 annotation. All reads mapping to protein coding genes were included in the differential expression analysis, which was performed using DESeq2 with normal log fold change shrinkage and taking into consideration pairing of samples among replicates (*Love et al., 2014*). For cumulative distribution plots $log_2$ fold change values derived from DESeq2 were used based on three biological replicates.

Raw fastq files of TRAP RNA-Seq from *Ceolin et al. (2017)* were downloaded from GEO submission GSE94559 and processed in the same way. Similar to this study, the authors used the RiboTag mouse bred to the *Fmr1* KO to explore differences in CA1 neurons in Fragile X. In contrast to our study they used the *Wfs1-CreERT2* mouse line and induced recombination by administration of tamoxifen in adult animals (2–6 months old).

For comparison of TRAP and CLIP data, WT TRAP reads were also aligned to the mm10 RefSeq transcriptome using STAR with quantMode GeneCounts TranscriptomeSAM. RPKM was determined for each transcript and a single transcript with the highest mean RPKM across all WT replicates was selected per gene. Having reduced the transcriptome to a single transcript per gene the RPKM values were recalculated and the transcriptome was further filtered for RPKM $\geq$ 1. This set of 10883 transcripts was used for all subsequent analysis.

### CLIP read processing and alignment

CLIP reads were processed as described previously using CLIP Tool Kit (CTK) (*Moore et al., 2014*; *Shah et al., 2017*). Briefly, raw reads were filtered for quality and demultiplexed using indexes introduced during the reverse transcription reaction. PCR duplicates were collapsed and adapter sequences removed. Reads were mapped to the mm10 RefSeq genome and transcriptome using STAR with quantMode GeneCounts TranscriptomeSAM. Mapped reads were further collapsed for potential PCR duplicates by coordinates and taking into consideration the degenerate barcodes introduced during the reverse transcription. After collapsing the transcriptome down to a single transcript per gene based on TRAP (see above), only reads that were uniquely mapped in the appropriate direction were retained.

## CLIP read distribution

The distribution of CLIP reads that uniquely mapped to the genome was determined using read_distribution.py from the RSeQC package (*Wang et al., 2012*) and GENCODE annotation.

For analysis of distribution across individual transcripts, reads from all biological replicates were combined. The reads were converted into a GRanges object using the GenomicRanges R package and coverage computed using the coverage function (*Lawrence et al., 2013*).

For meta-transcript analysis of read distribution relative to the start and stop codons, reads from all biological replicates were combined and the 1000 transcripts with highest CLIP tag density across the entire transcript were selected. Only reads that mapped to these transcripts were included in the analysis. Overall 67% of all reads mapped to this subset of transcripts. Each read was assigned a position relative to the start and stop codon based on the center of the read and reads were then selected that fell within the region of interest, for example 1 kb upstream or downstream of the stop codon. The total number of reads within each 10 nt window across the region was calculated. To ensure equal representation of the 1000 transcripts in the meta-transcript representation each read was weighted according to the total number of reads that mapped to that transcript within the region of interest. The data was further normalized for the proportion of the 1000 transcripts that were represented within that window since not all transcripts had a sufficiently long coding region or UTR to cover the full region of interest.

## CLIP score calculation

CLIP scores were calculated for each biological replicate individually. For each transcript the total number of reads which overlapped with the coding region was calculated for both the *Fmr1*-cTag-$^{Camk2a-Cre}$ samples (CrePos counts) and Cre negative *Fmr1*-cTag controls (CreNeg counts). *Fmr1*-cTag$^{Camk2a-Cre}$ samples and parallel Cre negative *Fmr1*-cTag control samples were indexed at the reverse transcription step of the library preparation and pooled immediately after, such that both DNA libraries were purified and amplified together. Addition of a degenerate sequence in the RT primers enabled collapse of PCR duplicates. These steps enabled the size of the *Fmr1*-cTag$^{Camk2a-Cre}$ and *Fmr1*-cTag control libraries to have differing read depths that were relative to the number of CLIP tags isolated in the initial immunoprecipitation.

The coding region RPKM for a specific transcript i was calculated as:

$$RPKM_i = \frac{(CrePosgtCounts_i - CreNeggtCounts_i) \times 10^9}{CDSgtLength \times (TotalgtCrePosgtCounts - TotalgtCreNeggtCounts)}$$

For any transcripts that had more reads in the Cre negative control than the *Fmr1*-cTag$^{Camk2a-Cre}$ CLIP or no reads in either sample the RPKM was set to 1.

Scatter plots of $log_2$ CLIP RPKM vs $log_2$ WT TRAP RPKM were plotted for all transcripts with at least one coding region CLIP tag and a linear regression line fitted to the data. The CLIP score for each transcript i and each replicate j was determined as the difference between the observed CLIP RPKM and the fitted value for the relevant TRAP RPKM expression level and calculated from the slope (lm slope) and intercept (lm intercept) of the fitted line:

$$CLIPgtscore_{ij} = log_2gtCLIPgtRPKM_{ij} - (lmgtslope \times log_2gtTRAPgtRPKM_i) + lmgtintercept$$

CLIP scores were calculated for all transcripts regardless of whether they had any CLIP tags. The final CLIP score for each transcript was determined as the mean CLIP score across all biological replicates.

Transcripts were classed as stringent targets if they had a CLIP score $\geq 2$ across all biological replicates, high binding targets if they had a CLIP score $\geq 1$ but did not meet the criteria for stringent target and low binding targets if they had a CLIP score between 0 and 1. All other transcripts were classed as non-targets. When a single target group was required for analysis purposes stringent and high binding targets were combined but low binding targets were not included.

## Alternative methods for normalizing CLIP

Normalization of FMRP CLIP data to transcript abundance was also tested with two alternative methods for comparison. Firstly, the same linear regression method as above was calculated but

substituting TRAP RPKM per transcript for FACS RNA-Seq RPKM values as a measure of transcript abundance.

Secondly, we used raw counts rather than RPKM values to compare CLIP and TRAP data. This alternative approach incorporated a dispersion estimate and used a negative binomial distribution to determine significance. Counts per CDS region were determined from CLIP and TRAP replicates. For CLIP counts CreNeg counts were subtracted from CrePos counts as before to give the final CLIP counts used in the analysis. All transcripts with at least one CLIP count across the 3 CLIP replicates were included. We modelled the FMRP CLIP read counts driven by mRNA abundance as following a negative binomial distribution with mean $\mu$ and dispersion $\alpha$. We estimated the dispersion parameter $\alpha$ for any $\mu$ by leveraging the observed per-transcript CLIP count variability across replicates using the approach implemented in DESeq2 (i.e. estimateDispersions). For each replicate, the expected CLIP count $\mu$ for each transcript based on mRNA abundance was estimated by linear regression that was fitted to the logarithm of the CLIP CDS vs TRAP CDS counts. CLIP scores were determined as before based on residuals from the fitted line. In addition, the significance of deviation of the observed CLIP count compared to the expected count ($\mu$) was calculated as a one-sided p-value for the negative binomial distribution, using the estimated dispersion parameter ($\alpha$). P-values from individual replicates were combined using Fisher's method and converted to false discovery rate (FDR) using the Benjamini and Hochberg method.

## Comparison of CLIP across cell types

For comparison of CLIP scores between CA1 neurons and cerebellar granule cells, only transcripts which had at least one CLIP tag in every biological replicate across both cell types were included as these are the transcripts for which the most accurate CLIP score can be determined. The limma R package (*Ritchie et al., 2015*) was used to compare CLIP scores from three biological replicates of FMRP cTag CLIP from CA1 neurons generated using the Camk2a Cre driver and three biological replicates of FMRP cTag CLIP from granule cells using the NeuroD1 Cre driver (*Van Driesche et al., 2019*). Transcripts were considered to be differentially bound by FMRP if they had an adjusted p-value≤0.05 and a mean CLIP score $\geq$ 0 in at least one of the cell types.

## ASD gene scores

The SFARI gene dataset with gene scores was downloaded from SFARI Gene database at gene.sfari.org. ASD gene associations have also been determined by using network-neighborhood differential enrichment analysis of de novo coding and non-coding mutations from whole genome sequencing of 1790 ASD simplex families (Supplementary file 3 from *Zhou et al., 2019*). This approach identifies genes whose functional network neighborhood is significantly enriched for genes with stronger predicted disease impact in proband mutations as compared to sibling mutations. Z scores generated from the p-values from this analysis were used as an ASD gene score. For comparison of ASD gene scores of FMRP targets between cell types, only transcripts that were expressed in both cell types were considered. ASD gene scores for all FMRP targets with a CLIP score $\geq$ 1 were compared between cell types.

## GO term analysis

Gene ontology (GO) enrichment analysis was performed using GOrilla (cbl-gorilla.cs.technion.ac.il; *Eden et al., 2009*). Ranked lists based on CLIP score or $\log_2$ fold change between KO and WT TRAP were used and GO terms that were significantly enriched at the top of the ranked list of genes were identified.

## GSEA analysis

Gene Set Enrichment Analysis (GSEA) was performed using the GSEA software from the Broad Institute using the GSEApreranked algorithm and the C2 curated gene sets from the Molecular Signatures Database (*Liberzon et al., 2011*; *Mootha et al., 2003*; *Subramanian et al., 2005*). Ranked lists based on CLIP score or $\log_2$ fold change between KO and WT TRAP were used and gene sets that were significantly enriched at either end of the ranked list of genes were identified. Alternatively, for analysis of enrichment/depletion of selected cell type-specific markers, selected gene sets were compared to a ranked list of log2 fold change between TRAP and Input samples. Gene

markers for each cell type were taken from Supplementary Table 1 from *Zeisel et al. (2015)* or were downloaded from DropViz.org (*Saunders et al., 2018*).

## Acknowledgements

We thank Darnell lab members for suggestions, expertise, and feedback. We thank Paula Cutrim for technical assistance. This work was supported by funds from the NIH to RBD (NS034389, NS081706, NS097404, and 1UM1HG008901) and to JCD (R01 HD040647) and the Simons Foundation to RBD (SFARI 240432). RBD is an Investigator of the Howard Hughes Medical Institute.

## Additional information

### Funding

| Funder | Grant reference number | Author |
| --- | --- | --- |
| National Institutes of Health | NS034389 | Robert B Darnell |
| National Institutes of Health | NS081706 | Robert B Darnell |
| National Institutes of Health | NS097404 | Robert B Darnell |
| National Institutes of Health | 1UM1HG008901 | Robert B Darnell |
| National Institutes of Health | R01 HD040647 | Jennifer C Darnell |
| Simons Foundation | SFARI 240432 | Robert B Darnell |
| Howard Hughes Medical Institute | | Robert B Darnell |

The funders had no role in study design, data collection and interpretation, or the decision to submit the work for publication.

### Author contributions

Kirsty Sawicka, Conceptualization, Data curation, Formal analysis, Validation, Investigation, Visualization, Methodology, Project administration; Caryn R Hale, Investigation, Methodology; Christopher Y Park, Formal analysis, Visualization, Methodology; John J Fak, Jin Joo Kang, Validation, Investigation; Jodi E Gresack, Resources, Provided critical advice and training for behavioral testing and analysis; Sarah J Van Driesche, Resources, Investigation; Jennifer C Darnell, Conceptualization, Supervision, Funding acquisition; Robert B Darnell, Conceptualization, Supervision, Funding acquisition, Project administration

### Author ORCIDs

Kirsty Sawicka (iD) https://orcid.org/0000-0003-4195-6327
Robert B Darnell (iD) https://orcid.org/0000-0002-5134-8088

### Ethics

Animal experimentation: Animals were treated in accordance with the principles and procedures of the National Institutes of Health Guide for the Care and Use of Laboratory Animals. All mouse experiments were approved by The Rockefeller University Institutional Animal Care and Use Committee (protocol number 17013).

### Decision letter and Author response

Decision letter https://doi.org/10.7554/eLife.46919.sa1
Author response https://doi.org/10.7554/eLife.46919.sa2

## Additional files

**Supplementary files**

• Supplementary file 1. Summary of CA1 FMRP cTag CLIP read alignment. Summary of read alignment for each replicate of FMRP cTag CA1 CLIP including *Fmr1*-cTag[Camk2a-Cre] and *Fmr1*-cTag negative control CLIP. The total number of reads obtained after demultiplexing and initial collapse of exact PCR duplicates is given ('input reads for alignment') as well as the number of these reads which uniquely mapped to the genome ('Uniquely mapped to genome') and transcriptome ('Mapped within transcriptome and collapsed for PCR duplicates'). Finally, the number of reads which mapped to the positive strand of the CA1 transcriptome (as defined by CA1 TRAP rpkm > 1) is shown ('Mapped within CA1 TRAP defined transcriptome for CLIP score determination').

• Supplementary file 2. Summary of CA1 FMRP cTag CLIP and granule cell. FMRP cTag CLIP data including counts per transcript, CLIP scores, FMRP target classifications and CA1 or granule cell-enriched binding. *CA1 CLIP*. CLIP tags that mapped within the coding region of each transcript were counted for each CLIP replicate and the final read count calculated by subtracting the number of CLIP reads in the Cre negative control from the number of CLIP reads from the Camk2a-Cre sample (CDS counts). These counts were converted to RPKM based on the length of the coding region. After normalization to abundance using TRAP RPKM per transcript as a proxy, the CLIP score for the transcript was calculated. *CA1 Targets*. List of transcripts classified as stringent, high binding or low-binding FMRP Targets in CA1 neurons. Stringent targets have a CLIP score > 2 in all three replicates, high binding targets have a mean CLIP score > 1 and low binding targets have a mean CLIP score between 0 and 1. *Granule Cell CLIP*. FMRP cTag CLIP was performed from cerebellar granule cells using Neurod1-Cre. Table of transcript information and CLIP scores is reproduced from *Van Driesche et al. (2019)*. *Granule Cell Targets*. List of transcripts classified as stringent, high binding or low-binding FMRP Targets in cerebellar granule cells. Stringent targets have a CLIP score > 2 in all three replicates, high binding targets have a mean CLIP score > 1 and low binding targets have a mean CLIP score between 0 and 1. *Enriched CA1*. Transcripts with significantly higher binding in CA1 compared to cerebellar granule cells. Differential CLIP scores were determined using Limma. $Log_2$ fold change and p-values are reported. *Enriched Granule Cell*. Transcripts with significantly higher binding in cerebellar granule cells compared to CA1 neurons. Differential CLIP scores were determined using Limma. $Log_2$ fold change and p-values are reported.

• Supplementary file 3. Differential expression analysis of *Fmr1* KO vs WT TRAP from CA1 neurons and cerebellar granule cells. Results from *Fmr1* KO vs WT TRAP. Raw counts per gene and statistics from DESeq2 analysis of differential expression between WT and KO are shown. *Sawicka CA1 Fmr1 KO TRAP DESeq2*. CA1 TRAP data from RiboTag[Camk2a-Cre] P28-P31 mice presented in this study. *Ceolin CA1 Fmr1 KO TRAP DESeq2*. CA1 TRAP data from RiboTag[Wfs1-CreERT2] 2–6 month old mice published by *Ceolin et al. (2017)*. Downloaded from GSE94559. *Granule Fmr1 KO TRAP DESeq2*. Cerebellar granule cell TRAP data from RiboTag[Neurod1-Cre] 6–8 week mice presented in this study.

• Supplementary file 4. RT primers for CLIP Sequences for reverse transcription primers each containing a six nucleotide barcode index highlighted in red to allow pooling and multiplexing of samples for BrdU immunoprecipitation, PCR and sequencing.

• Supplementary file 5. Alternative CLIP normalization methods. *CLIP normalized to FACS RNA-Seq*. CLIP scores determined from linear model using FACS RNA-Seq transcript RPKM. This table is equivalent to *Supplementary file 2* CA1 CLIP but using RPKM per transcript from RNA-Seq of sorted CA1 pyramidal cells to normalize for abundance instead of TRAP. *Counts and negative binomial*. FMRP targets determined using a count-based method in which the FMRP CLIP read counts driven by mRNA abundance are modelled as following a negative binomial distribution. Reads from CLIP and TRAP replicates that mapped within the coding region of each transcript were counted. For CLIP the final read count was calculated by subtracting the number of CLIP reads in the Cre negative control ('Neg CLIP CDS reads') from the number of CLIP reads from the Camk2a-Cre sample ('Camk2a CLIP CDS reads'). The mean TRAP counts normalized by library size across all three replicates are shown ('mean normalized TRAP counts'). CLIP scores were derived from a linear regression model as the difference between the observed and fitted values. An estimated dispersion parameter

for each CLIP count is given, along with individual p-values per replicate, combined p-value and FDR per transcript.

• Transparent reporting form

## Data availability

Sequencing data has been uploaded to GEO under the SuperSeries GSE127847 comprising Subseries GSE127845, GSE127846 and GSE135147.

The following datasets were generated:

| Author(s) | Year | Dataset title | Dataset URL | Database and Identifier |
|---|---|---|---|---|
| Sawicka K, Darnell JC, Darnell RB | 2019 | TRAP from CA1 pyramidal neurons and cerebellar granule cells in Fmr1 KO and wildtype mice | https://www.ncbi.nlm. nih.gov/geo/query/acc. cgi?acc=GSE127845 | NCBI Gene Expression Omnibus, GSE127845 |
| Sawicka K, Hale CR, Darnell JC, Darnell RB | 2019 | FMRP cTag CLIP from mouse CA1 pyramidal neurons | https://www.ncbi.nlm. nih.gov/geo/query/acc. cgi?acc=GSE127846 | NCBI Gene Expression Omnibus, GSE127846 |
| Sawicka K, Darnell JC, Darnell RB | 2019 | RNA-Seq from CA1 pyramidal neurons | https://www.ncbi.nlm. nih.gov/geo/query/acc. cgi?acc=GSE135147 | NCBI Gene Expression Omnibus, GSE135147 |

The following previously published dataset was used:

| Author(s) | Year | Dataset title | Dataset URL | Database and Identifier |
|---|---|---|---|---|
| Ceolin L, Bouquier N, Vitre J, Bertaso F, Rialle S, Severac D, Valjent E, Puighermanal E, Perroy J | 2017 | Hippocampus CA1 pyramidal cells Transcriptomic profile in WT and Fmr1 KO mice, using Wfs1-CreERT2:RiboTag:Frm1 knockout and wildtype mice | https://www.ncbi.nlm. nih.gov/geo/query/acc. cgi?acc=GSE94559 | NCBI Gene Expression Omnibus, GSE94559 |

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
