## [Decision Letter]

Thank you for submitting your article "FMRP has a cell-type-specific role in CA1 pyramidal neurons to regulate autism-related transcripts and circadian memory" for consideration by *eLife*. Your article has been reviewed by Huda Zoghbi as the Senior Editor, a Reviewing Editor, and three reviewers. The reviewers have opted to remain anonymous.

The reviewers have discussed the reviews with one another and the Reviewing Editor has drafted this decision to help you prepare a revised submission.

Summary:

This study aims to identify neuronal cell-type specific functional roles of FMRP with implications in FXS pathophysiology. The authors take advantage of their elegant cTag CLIP approach to endogenously tag FMRP with AcGFP in CA1 pyramidal neurons or cerebellar granule cells and perform CLIP-Seq to determine FMRP direct targets in these cell types in the mouse brain. The cTag CLIP-seq data are compared with TRAP data from the same cell types in WT and Fmr1 deficient mice to assess effects of loss of FMRP binding on gene expression. Analysis of these data resulted in several interesting observations including that FMRP stabilizes targets and that FMRP targets are enriched for autism- as well as circadian-related transcripts, particularly in CA1 pyramidal neurons. The authors complement these observations with behavioural analyses revealing that FMRP deficiency has differential effects on memory depending on the state of the circadian cycle. Overall, this is an interesting study that provides timely cell type-specific insights into FMRP function in vivo. However, there are several technical concerns and additional comments that the authors are requested to address before their manuscript can be considered further.

Essential revisions:

1) The authors' approach of using CA1-specific TRAP data may be problematic for assessing transcript abundance for the purpose of normalizing FMRP-binding. This is especially a concern given evidence of reduced ribosome occupancy over transcripts bound with high affinity by FMRP (e.g. Sharma et al., 2019). We recommend that the authors perform the following experiment to address this concern, or at the very least more clearly explain the caveats/justification of their approach and how it may affect interpretations.

Perform FACS of CA1 labelled neurons, followed by RNA-seq to assess transcript abundance and use these data to normalize FMRP cTag CLIP signal. A significant correlation between FMRP targets detected by this approach and those detected using TRAP data would provide more confidence in the current results.

2) The Cre-lox dependent tagging of FMR1 could affect downstream interpretation since Fmr1 harbours alternative last coding exons and other processing events (eg. Zimmer et al., 2016; Tseng et al., 2017). It is recommended that the authors provide RT-PCR and western evidence from FMRP-tagged as well as wild type animals to show that the Cre-lox tagging strategy captures the majority of FMRP isoforms.

3) Figure 1B. Related to (2), it is recommended that the authors perform immunofluorescence microscopy with an antibody against FMRP to confirm whether the GFP tagged protein co-localizes with endogenous protein.

4) The authors use unpublished data (cited as Van Driesche et al., in submission) in their manuscript, yet the corresponding methods are not described, and their paper is not available in BioRxiv. As such, analyses involving these data (especially in the section "Comparison of CLIP across cell types") are difficult to interpret and assess. The authors are requested to provide a more detailed description of these data and methods, and their paper should be made available to the reviewers. Related to this, the reviewers have several questions/concerns about the data analysis which the authors are requested to address.

5) In the CLIP score calculation, the formula for RPKM used by the authors (which is somewhat unclear because of how it is typed out in the methods section) appears to be as follows:

〖RPKM〗_i = ((〖CrePos Counts〗_i-〖CreNeg Counts〗_i)(1000000)) / ((L/1000)(CrePos Total Counts-Cre Neg Total Counts))

where i is a specific transcript, L is the CDS length, and all counts originate in the CDS. It is recommended that Equation Editor is used to write the equation, but there is no way to do this in the review submission site, so just for clarity:

Numerator = (〖CrePos Counts〗_i-〖CreNeg Counts〗_i)(1000000)

Denominator = ((L/1000)(CrePos Total Counts-Cre Neg Total Counts))

This formula seems strange because if the Cre+ and Cre- samples are sequenced to the same depth (which is probably what the authors would attempt to do), then the denominator would be zero. I'm guessing that the actual formula probably comprises two separately normalized terms for the Cre+ and Cre- counts, but I could be misunderstanding something. Please clarify.

6) The authors use a linear regression model to identify high-affinity binding targets for FMRP based on their CLIP score analysis. This seems problematic because the data used as input are counting data, and this analysis does not appear to take counting noise and overdispersion into account. It is also unclear whether any statistical test was applied to this analysis. In order to compute the cumulative histograms in, for example, Figure 2, the authors used the generalized linear model in DESeq2 and applied it in a sophisticated way that takes sample pairing into account. Why not apply this same approach to the CLIP analysis (after taking into consideration the issues with normalization raised in point 1)? In principle, DESeq2 will account for coverage differences between samples and genes, account for overdispersion, and provide a test statistic for each transcript.

7) The authors show a handful of cell type-specific transcripts on a volcano plot in Figure 2A to demonstrate the cell type-specificity of their TRAP experiments. The authors should consider an unbiased and systematic assessment with larger gene sets using gene set enrichment analysis (GSEA), which the authors deployed in other parts of the paper. There are now numerous sources of cell type-specific gene sets for various neuronal and glial cell types from both bulk RNA-seq (e.g. from Barres and colleagues) and single-cell RNA-seq (e.g. from Linnarsson, McCarroll, and others).

8) The authors are requested to clarify how they controlled for possible length biases in their calculations. This is especially important when determining the validity of the observations in Figure 3E. The use of RPKM is valid as long as transcript length is stable (i.e. FMRP is not affecting transcript length).

9) In the section investigating FMRP regulation of circadian rhythm transcripts, the authors should describe the background and cutoffs used in their functional analysis. This section would be strengthened if the authors could address whether FMRP affects the circadian-dependent changes in the expression of these genes by performing RT-(q)PCR and western blotting (as they have in Figure 2D-F).

10) At present it is unclear whether the circadian genes regulated by FMRP are only expressed and/or bound in CA1 or whether such binding occurs in other brain regions important for circadian regulation. It is also unclear whether there are alterations in the rhythmic expression of clock genes in the hippocampus of Fmr1 KO mutants (related to point 9). At present, evidence relies heavily on the behavioural read-outs reported in the manuscript, which can be confounded by various factors affecting behaviours that are independent of FMRP's role in the regulation of circadian genes in the CA1 region. These caveats should be discussed.

---

## [Author Response]

Essential revisions:1) The authors' approach of using CA1-specific TRAP data may be problematic for assessing transcript abundance for the purpose of normalizing FMRP-binding. This is especially a concern given evidence of reduced ribosome occupancy over transcripts bound with high affinity by FMRP (e.g. Sharma et al., 2019). We recommend that the authors perform the following experiment to address this concern, or at the very least more clearly explain the caveats/justification of their approach and how it may affect interpretations.Perform FACS of CA1 labelled neurons, followed by RNA-seq to assess transcript abundance and use these data to normalize FMRP cTag CLIP signal. A significant correlation between FMRP targets detected by this approach and those detected using TRAP data would provide more confidence in the current results.

Thank you for your valuable suggestion. We acknowledge that TRAP will only enable isolation of mRNAs that are ribosome associated and therefore is not a perfect measure of transcript abundance. In optimizing our TRAP protocol we have taken care to titrate the antibody to maximize depletion of the ribosome-associated pool of mRNAs (Figure 2—figure supplement 1) and thereby capture as best as possible even those transcripts with low numbers of associated ribosomes.

To confirm that our TRAP data is representative of the transcriptome, we have taken the reviewer’s suggestion and used FACS to isolate CA1 neurons labelled in vivo with tdTomato (Figure 2—figure supplement 2). While this technique is also not perfect for defining the in vivo transcriptome (due to the potential for changes to the transcriptome during the dissociation and sorting process and the loss of neuronal processes), we agree with the reviewers that concordance of the FMRP targets defined using these two different measures of transcript abundance provides more confidence in our findings.

We found that TRAP and FACS sequencing data were remarkably similar, with a correlation of R^2^ = 0.87 (Figure 2—figure supplement 2B). Those genes that were found to be differentially expressed between these two methods, are easily interpretable. For instance, histone transcripts were enriched in TRAP vs FACS since poly(A) selection was used in the preparation of FACS RNA-seq data and histone transcripts are not polyadenylated. In contrast, transcripts from the mitochondrial genome were only observed in FACS since these are translated by the mitochondrial ribosome which does not contain the labelled RPL22 ribosomal protein. Other transcripts that were enriched in TRAP vs FACS, include those encoding synaptic proteins and mitochondrial proteins (from the nuclear genome) that may be locally translated in neuronal processes removed during tissue dissociation.

We also compared CLIP scores determined by normalization to FACS RNA-seq to those calculated by normalization to TRAP as the measure of RNA abundance. Consistent with the high correlation between the read counts obtained with the two methods, the CLIP scores are also highly correlated (Figure 2—figure supplement 4A). Furthermore, the set of stringent targets defined in both cases showed the same decrease in CA1 TRAP in *Fmr1* KO mice (Figure 2—figure supplement 4C).

2) The Cre-lox dependent tagging of FMR1 could affect downstream interpretation since Fmr1 harbours alternative last coding exons and other processing events (eg. Zimmer et al., 2016; Tseng et al., 2017). It is recommended that the authors provide RT-PCR and western evidence from FMRP-tagged as well as wild type animals to show that the Cre-lox tagging strategy captures the majority of FMRP isoforms.

The reviewers raise an excellent point and it’s an issue that we’re aware of and have addressed in the paper now available on BioRxiv that describes the generation of the *Fmr1-*cTag mouse and its characterization/validation (Van Driesche et al., 2019).

There is a large literature on alternative splicing of the mouse and human *FMR1* pre-mRNA to generate multiple protein isoforms starting with the first description of alternative splicing by David Nelson (Verkerk et al., 1993) and including the two papers cited by the reviewers from Mike Akins and Flora Tassone’s labs (Brackett et al., 2013; Dury et al., 2013; Fu et al., 2015; Huang et al., 1996; Sittler et al., 1996; Tseng et al., 2017; Zhang et al., 2019; Zimmer et al., 2017). These publications all agree that by far the most common splice variants are (1) the alternative splicing of exon 12, and (2) alternate splice acceptor sites in exons 15 (two) and 17 (one). However, a very small percentage of total FMRP in brain (perhaps 4-6%) is present as nuclear isoforms resulting from the exclusion of exon 14 (Brackett et al., 2013; Fu et al., 2015). This is significant for our work because it results in an alternate C-terminus on the FMRP protein. Our C-terminal tagging strategy will miss these FMRP isoforms because of the novel C-termini. In the paper describing the *Fmr1*-cTag mouse, we acknowledge and discuss that we are unable to query the RNA binding properties of these isoforms and we have added a statement to the current manuscript to make it clear that we are only assaying the predominant isoforms with the usual C-terminus but missing the minority which splice exon 14 out.

Subsection “FMRP cTag CLIP in CA1 neurons”: “The addition of the AcGFP-tag.… enables capture of all major isoforms of FMRP with the exception of a small minority of nuclear isoforms with an alternative c-terminus resulting from exclusion of exon 14”

Of relevance, the antibodies we used in the past for native FMRP CLIP were all generated against C-terminal epitopes (to avoid immunoprecipitating FXR1 and FXR2 which are very homologous through exon 14) and so they also fail to capture FMRP isoforms with alternate C-termini. While it’s unfortunate that we can’t capture all isoforms of FMRP with the cTag approach we are confident that the isoforms that we are able to capture make up the vast majority of FMRP in neurons and those associated with polyribosomes, and that the current data adds to our understanding of important functions of FMRP in specific cell types.

In the paper describing the generation of the cTag mouse we used RT-PCR to amplify sequences from exon 11 to the native 3’UTR in exon 17 in Cre(-) littermates or from exon 11 to the novel GFP tag in new exon 17 following Cre-dependent recombination (see Author response image 1, which is data from Van Driesche et al., 2019). We found a very similar pattern of isoforms in both cases. We also used Western blotting to detect FMRP bands after running SDS-PAGE gels longer than usual to resolve the higher MW isoforms containing the GFP tag. Here we found that FMRP resolved into additional isoforms that appear similar in ratio to the endogenous forms, but cannot be quantitative about the identity or abundance of each (please see Author response image 1).

In summary, we realize we are missing a potentially important group of nuclear FMRP isoforms and acknowledge this in both papers. It will be pursued using antibodies specific to the novel C-termini in the future.

3) Figure 1B. Related to (2), it is recommended that the authors perform immunofluorescence microscopy with an antibody against FMRP to confirm whether the GFP tagged protein co-localizes with endogenous protein.

Unfortunately, since FMRP is encoded on the X-chromosome and we have inserted the GFP tag at the endogenous locus, it isn’t possible to have the tagged version of the protein and the endogenous protein expressed in the same cell. The distribution of the GFP-tagged protein is however consistent with that observed for FMRP in wild-type animals both by us (please see Author response image 2) and others (Christie et al., 2009; Davidovic et al., 2006).

**Author response image 2. respfig2:** 

We have performed extensive validation to confirm that the addition of the GFP tag does not change other properties of FMRP, including association with polysomes or mRNA targets. We apologize that this data wasn’t available to reviewers during the initial review process but have now uploaded the manuscript containing the full validation of the FMRP cTag mouse model to BioRXiv (Van Driesche et al., 2019). A copy of the manuscript has also been included in the resubmission of the current manuscript for the reviewers.

4) The authors use unpublished data (cited as Van Driesche et al., in submission) in their manuscript, yet the corresponding methods are not described, and their paper is not available in BioRxiv. As such, analyses involving these data (especially in the section "Comparison of CLIP across cell types") are difficult to interpret and assess. The authors are requested to provide a more detailed description of these data and methods, and their paper should be made available to the reviewers. Related to this, the reviewers have several questions/concerns about the data analysis which the authors are requested to address.

Again, we apologize that this data wasn’t available to reviewers during the initial review process. The manuscript containing a full description and analysis of FMRP cTag CLIP in cerebellar neurons has been uploaded to BioRXiv (Van Driesche et al., 2019) and a copy of the manuscript has also been included in the resubmission of the current manuscript for the reviewers.

5) In the CLIP score calculation, the formula for RPKM used by the authors (which is somewhat unclear because of how it is typed out in the methods section) appears to be as follows:〖RPKM〗_i = ((〖CrePos Counts〗_i-〖CreNeg Counts〗_i)(1000000)) / ((L/1000)(CrePos Total Counts-Cre Neg Total Counts))where i is a specific transcript, L is the CDS length, and all counts originate in the CDS. It is recommended that Equation Editor is used to write the equation, but there is no way to do this in the review submission site, so just for clarity:Numerator = (〖CrePos Counts〗_i-〖CreNeg Counts〗_i)(1000000)Denominator = ((L/1000)(CrePos Total Counts-Cre Neg Total Counts))This formula seems strange because if the Cre+ and Cre- samples are sequenced to the same depth (which is probably what the authors would attempt to do), then the denominator would be zero. I'm guessing that the actual formula probably comprises two separately normalized terms for the Cre+ and Cre- counts, but I could be misunderstanding something. Please clarify.

Thank you for your comments and we hope that the revised description of the CLIP score calculation provided in the methods is now clearer. In addition to using an equation editor to display the equations, we have also added further explanation in the methods section which describes how our library preparation methods enables the sequencing depths of the Cre positive and Cre negative samples to be representative of the number of RNA fragments immunoprecipitated in each case.

Subsection “CLIP score calculation”: “Fmr1-cTag^Camk2a-Cre^ samples and parallel Cre negative Fmr1-cTag control samples were indexed at the reverse transcription step of the library preparation and pooled immediately after, such that both DNA libraries were purified and amplified together. Addition of a degenerate sequence in the RT primers enabled collapse of PCR duplicates. These steps enabled the size of the Fmr1-cTag^Camk2a-Cre^and Fmr1-cTag control libraries to have differing read depths that were relative to the number of CLIP tags isolated in the initial immunoprecipitation.

The coding region RPKM for a specific transcript i was calculated as:RPKMi=(CrePoscountsi−CreNegCountsi)×109CDSLength×(TotalCrePoscounts -TotalCreNegCounts)

6) The authors use a linear regression model to identify high-affinity binding targets for FMRP based on their CLIP score analysis. This seems problematic because the data used as input are counting data, and this analysis does not appear to take counting noise and overdispersion into account. It is also unclear whether any statistical test was applied to this analysis. In order to compute the cumulative histograms in, for example, Figure 2, the authors used the generalized linear model in DESeq2 and applied it in a sophisticated way that takes sample pairing into account. Why not apply this same approach to the CLIP analysis (after taking into consideration the issues with normalization raised in point 1)? In principle, DESeq2 will account for coverage differences between samples and genes, account for overdispersion, and provide a test statistic for each transcript.

In response to the valid concerns raised about the linear regression model used to determine the CLIP score, we have explored different ways in which to determine FMRP targets from our data.

Whilst DESeq2 is a powerful method for the comparison of RNA-Seq datasets, we feel that it is unsuitable for comparison of CLIP and RNA-seq which have very different depths, noise and dispersion, since DESeq2 applies a single dispersion estimate across all samples. Instead of using a standard DESeq2 analysis pipeline for the comparison, we have established an alternative analysis method that uses the raw counts across the coding region as the input, makes use of some of the elements implemented within the DESeq2 package in conjunction with a linear regression method and uses a negative binomial distribution to determine significance. This enables incorporation of a dispersion estimate and a statistical test. For further details please see a full description within the updated methods section (subsection “Alternative methods for normalizing CLIP”).

Stringent targets, as defined by this alternative method and an adjusted p-value < 0.05, were similar to those identified by our original linear regression model and showed an equal decrease in abundance in *Fmr1* KO TRAP (Figure 2—figure supplement 4).

Ultimately, given that none of the methods we have tested (use of FACS RNA-Seq data instead of TRAP or analysis incorporating a dispersion estimate and using a negative binomial distribution) seem to out-perform any of the others, we have chosen to keep with the original analysis pipeline. This has the advantage of using RPKM rather than direct count data for the regression analysis which should reduce length bias in our calculations. Furthermore, we find that mRNAs encoding synaptic proteins are more highly represented in TRAP than FACS and therefore normalization to FACS RNA-Seq could potentially bias towards these locally translated mRNAs in our CLIP analysis. For transparency, we have included in the revised manuscript a description of the individual methods (subsection “Alternative methods for normalizing CLIP”), comparisons of the different methods (Figure 2—figure supplement 4) and a full table of results for each method as Supplementary file 5.

7) The authors show a handful of cell type-specific transcripts on a volcano plot in Figure 2A to demonstrate the cell type-specificity of their TRAP experiments. The authors should consider an unbiased and systematic assessment with larger gene sets using gene set enrichment analysis (GSEA), which the authors deployed in other parts of the paper. There are now numerous sources of cell type-specific gene sets for various neuronal and glial cell types from both bulk RNA-seq (e.g. from Barres and colleagues) and single-cell RNA-seq (e.g. from Linnarsson, McCarroll, and others).

As suggested, we have performed the proposed GSEA with two available single-cell RNA-Seq datasets from mouse hippocampus, one from the Linnarsson lab (Zeisel et al., 2015) and one from the McCarroll lab (Saunders et al., 2018). These corroborate our findings with a smaller subset of cell type specific markers and demonstrate enrichment of CA1 pyramidal cell-specific genes and depletion of markers of other hippocampal cell types in our immunoprecipitation. These data are provided in Figure 2—figure supplement 2A.

8) The authors are requested to clarify how they controlled for possible length biases in their calculations. This is especially important when determining the validity of the observations in Figure 3E. The use of RPKM is valid as long as transcript length is stable (i.e. FMRP is not affecting transcript length).

We thank the reviewers for this perceptive comment. Although we cannot completely rule out some length bias in our data (as with all sequencing-based datasets), we have taken all possible measures to control and test for this. As the reviewer noted, we have used RPKM rather than raw count data for our analysis to minimise the influence of transcript or coding region length in our analysis – something that we consider to be an important difference and improvement over our previously published FMRP whole brain CLIP target list. To confirm that transcript length is stable, we ran DEXSeq analysis on our WT and KO TRAP datasets and found no evidence for a change in exon usage or isoform expression in the *Fmr1* KO mouse.

Given the correlation we observe between transcript length and FMRP binding, we have confirmed that our identification of targets is not driven by length alone, something that could imply a source of bias. Towards this end, we examined whether the decrease observed in the *Fmr1* KO TRAP was specific to our identified targets based on CLIP score or was seen across all long transcripts. We took a randomly selected length-matched set of control transcripts that had a CLIP score < 1 and compared them to our stringent FMRP targets. In this analysis, no difference in the *Fmr1* KO was observed for the length-matched control transcripts, only the identified FMRP targets (Figure 3F). This provides strong evidence that we are selecting a specific subset of FMRP-regulated long transcripts in our analysis. We have also confirmed that our identification of targets is not biased towards highly expressed genes using the same approach by comparing stringent targets to a randomly selected expression-matched set of control transcripts. Again only the targets were found to be down-regulated in the *Fmr1* KO (Figure 3—figure supplement 1E).

We have also considered whether 5’ or 3’ sequencing bias in our TRAP data could have led to an underestimation of the abundance of long transcripts and thereby an over-representation of long transcripts among our FMRP targets after normalization. To assess this, we compared CLIP scores calculated using only TRAP reads mapping within a 1 kb window at either the 5’ end or 3’ end of each transcript. CLIP scores were highly correlated regardless of whether transcript abundance was estimated from TRAP reads across the full transcript or either of these 1 kb windows (Figure 3—figure supplement 1C). Furthermore, the length distribution of stringent targets remained the same in each case (Figure 3—figure supplement 1D).

In summary, we have made every effort to control and test for length bias in our data but acknowledge that we cannot completely rule out an influence of sequencing bias in our calculations. As such, we have added a comment on length bias in the Discussion section of the manuscript to address the potential for length bias in our data and its possible implications on our findings.

Discussion section: “… we have made every attempt to minimize and test for length bias in our data, including the use of RPKM in our CLIP score determination, the use of parallel approaches (CLIP, TRAP and FACS RNA-Seq) and validation of our results by RT-qPCR. Thus, we think it is unlikely that sequencing bias has a significant effect on our determination of FMRP targets, although it is impossible to completely rule out any such effect, which could in turn lead to some influence on our observation that FMRP targets are enriched for long transcripts. Perhaps the most compelling evidence that our selection of targets is biologically relevant and not an artefact of length bias is the fact that the differences seen in the Fmr1 KO by TRAP are specific to targets even when comparing to length-matched controls (Figure 3F).”

9) In the section investigating FMRP regulation of circadian rhythm transcripts, the authors should describe the background and cutoffs used in their functional analysis. This section would be strengthened if the authors could address whether FMRP affects the circadian-dependent changes in the expression of these genes by performing RT-(q)PCR and western blotting (as they have in Figure 2D-F).

We apologise for any lack of clarity in our description of the functional analysis performed. We have supplied further explanation here and also when describing these results within the manuscript.

We identified circadian genes as targets of FMRP using the gene set enrichment analysis (GSEA) tool from the Broad Institute. This algorithm takes a user supplied ranked list of genes (in this case a list of CA1 transcripts ranked either according to their CLIP score, or their log fold change in *Fmr1* KO TRAP) and searches for gene sets that are statistically enriched at either end of the ranked list. The gene sets used are from the Molecular Signature Database (MSigDb), a publicly accessible collection of curated gene sets that is maintained by the GSEA team. In our plots, we have displayed two primary statistical outputs from the GSEA analysis, the normalized enrichment score (NES) and the false discovery rate (FDR). The NES reflects the degree to which a gene set is overrepresented at the top or bottom of a ranked list of genes normalized for differences in gene set sizes. The FDR is the estimated probability that a gene set with a given NES represents a false positive finding. Thus, in this type of analysis there is no specific background dataset since the analysis uses the complete ranked list. As a cutoff we considered only those gene sets that were statistically significantly enriched (FDR < 0.05) in either our CLIP or TRAP ranked data.

Subsection “FMRP regulated mRNAs are enriched in specific biological functions”: “… we performed Gene set enrichment analysis (GSEA) on our CLIP score ranked list and Fmr1 KO TRAP fold change ranked lists. This is a computational method that determines whether an a priori defined set of genes shows statistically significant, concordant differences between two biological states (e.g. bound vs unbound by CLIP or down-regulated vs up-regulated by TRAP). The algorithm takes a user supplied ranked list of genes and performs an unbiased search for gene sets that are statistically enriched at either end of the ranked list.”

Figure legends: “Normalized enrichment score (NES) reflects the degree to which a gene set is overrepresented at the top or bottom of a ranked list of genes normalized for differences in gene set sizes and the q-value (false discovery rate) is the estimated probability that a gene set with a given NES represents a false positive finding. Black lines represent the position within the ranked list of each gene in the gene set.”

As additional validation that circadian-related FMRP target mRNAs are down-regulated in the *Fmr1* KO mouse, we have performed RT-PCR using hippocampal tissue from *Fmr1* KO and WT littermates for three stringent FMRP targets, Ppargc1a, Npas2 and Ncoa2. As anticipated, all 3 were significantly decreased in abundance in the *Fmr1* KO mouse (Figure 4—figure supplement 2).

10) At present it is unclear whether the circadian genes regulated by FMRP are only expressed and/or bound in CA1 or whether such binding occurs in other brain regions important for circadian regulation. It is also unclear whether there are alterations in the rhythmic expression of clock genes in the hippocampus of Fmr1 KO mutants (related to point 9). At present, evidence relies heavily on the behavioural read-outs reported in the manuscript, which can be confounded by various factors affecting behaviours that are independent of FMRP's role in the regulation of circadian genes in the CA1 region. These caveats should be discussed.

The reviewers raise an excellent point. At present, we only have cell-type-specific CLIP data from neurons in the CA1 and cerebellum. Whilst, our data support that dysregulation of circadian genes occurs in CA1 but not cerebellar granule cells, we cannot rule out that FMRP regulates circadian transcripts in other brain regions not yet investigated. We have added a discussion of these caveats, as well as the possible influence of other brain regions on the behavioral phenotypes observed, and have highlighted the importance of future studies to investigate this further.

Discussion: “We find a specific regulation of circadian transcripts in CA1 pyramidal cells but not cerebellar granule cells, despite similar expression of these transcripts in both cell types (Figure 4D, Figure 4—figure supplement 1D). To date, since in vivo cell-type-specific identification of FMRP binding and analysis of downstream regulation has only been performed in CA1 pyramidal neurons and cerebellar granule cells, it remains unclear if dysregulation of circadian transcripts exists in other circadian-relevant cell types in the *Fmr1* KO mouse and whether this may also contribute to the behavioral phenotypes observed here. Nonetheless, the finding of cell-type specific circadian control in general, and additionally that such regulation can be FMRP-dependent, are important new observations.”

Discussion: “While our results combined with previous findings of others are suggestive of a direct link between altered circadian gene expression in the hippocampal CA1 region and altered CA1-dependent circadian behaviors, it remains to be tested whether other cell types or brain regions also contribute to these phenotypes. Future work, for example using CA1 specific deletion of FMRP, will be necessary to establish a direct link.”

To address the question of whether there are alterations in the rhythmic expression of clock genes in the CA1 of *Fmr1* KO mice, we have performed additional experiments with tissue collected at multiple time points across the circadian cycle. We chose to examine 4 circadian transcripts in CA1 tissue from *Fmr1* KO mice and their WT littermates that have been shown to have the greatest oscillations in human hippocampus across the circadian cycle. As anticipated, all showed significant differences in mRNA levels by RT-PCR across the circadian cycle. Of these, 3 showed a significant difference in expression in the *Fmr1* KO at least one time point while having comparable expression at other times during the circadian cycle (Figure 4—figure supplement 3). These data suggest a possible dysregulation of circadian oscillations of clock gene expression in CA1 in the absence of FMRP and provide a molecular correlate to our behavioral data.